# Proteogenomics analysis unveils a TFG-RET gene fusion and druggable targets in papillary thyroid carcinomas

Aswini Krishnan[1,11], Jean Berthelet[1,11], Emilie Renaud[1], Sebastian Rosigkeit[1], Ute Distler [2], Eric Stawiski[3], Jing Wang[3], Zora Modrusan[4], Marc Fiedler [5], Mariann Bienz [5], Stefan Tenzer [2], Arno Schad[6], Wilfried Roth[6], Bernd Thiede [7], Somasekar Seshagiri [4,8], Thomas J. Musholt[9] & Krishnaraj Rajalingam [1,10 ✉]

Papillary thyroid cancer (PTC) is the most common type of endocrine malignancy. By RNA-seq analysis, we identify a RET rearrangement in the tumour material of a patient who does not harbour any known RAS or BRAF mutations. This new gene fusion involves exons 1–4 from the 5′ end of the Trk fused Gene (TFG) fused to the 3′ end of RET tyrosine kinase leading to a TFG-RET fusion which transforms immortalized human thyroid cells in a kinase-dependent manner. TFG-RET oligomerises in a PB1 domain-dependent manner and oligo-merisation of TFG-RET is required for oncogenic transformation. Quantitative proteomic analysis reveals the upregulation of E3 Ubiquitin ligase HUWE1 and DUBs like USP9X and UBP7 in both tumor and metastatic lesions, which is further confirmed in additional patients. Expression of TFG-RET leads to the upregulation of HUWE1 and inhibition of HUWE1 significantly reduces RET-mediated oncogenesis.

[1] Cell Biology Unit, University Medical Center of the Johannes Gutenberg University Mainz, 55131 Mainz, Germany. [2] Institute of Immunology, University Medical Center of the Johannes Gutenberg University Mainz, 55131 Mainz, Germany. [3] MedGenome, Inc., Foster City, CA, USA. [4] Molecular Biology Department, Genentech, Inc., South San Francisco, CA, USA. [5] MRC Laboratory of Molecular Biology, Cambridge Biomedical Campus, Francis Crick Avenue, Cambridge CB2 0QH, UK. [6] Institute of Pathology, University Medical Center Mainz, Mainz, Germany. [7] Department of Biosciences, University of Oslo, 0371 Oslo, Norway. [8] SciGenom Research Foundation, Bangalore, India. [9] Endocrine Surgery Section, Department of General Visceral and Transplantation Surgery, University Medicine, Mainz, Germany. [10] University Cancer Center Mainz, University Medical Center, Mainz, Germany. [11] These authors contributed equally: Aswini Krishnan, Jean Berthelet ✉ email: krishna@uni-mainz.de

Papillary thyroid carcinoma (PTC) is one of the most common form of thyroid cancer. A recent survey by Surveillance, Epidemiology, and End Results (SEER) program estimates 53,990 new cases of thyroid cancer and 2060 deaths in the US, which would account for 3.1% of overall cancer cases and 0.3% of all cancer deaths in 2018[1,2]. Although the death rate from thyroid cancer is low, its incidence increased annually by 3.6% from 1974 to 2013 and it is the most rapidly increasing cancer in the US[3]. Thyroid cancer occurs three times more frequently in women than in men, though studies indicate that male patients are presented with more aggressive stages when diagnosed and have lower disease free survival as well as higher mortality[4,5]. Nearly 80% of all thyroid cancers are PTCs[6]. Other major types of thyroid cancer include follicular thyroid cancer (FTC), anaplastic thyroid cancer (ATC) and medullary thyroid cancer (MTC), and the distinction is primarily based on the cell of origin and tissue architecture[7]. Thus far, surgery and radioiodine treatment (RAI) represent the major therapeutic avenue for patients[8]. Even though patients with PTCs mostly have a good prognosis, recurrence and aggressive metastases have been found in an increasing number of patients in the last years[9]. Central lymph node (LN) metastases are often detected in 20–90% PTC patients and considered as a factor contributing to recurrence and morbidity in a study employing a large cohort of patients[10]. The *BRAF* mutation resulting in its highly kinase active protein form BRAFV600E and mutations in the *RAS* gene, especially *NRAS*, are the most common genetic alterations seen in PTC[7]. Apart from *BRAF* and *NRAS* mutations, common genetic alterations in PTCs include gene fusions involving the *RET* gene giving rise to oncogenic fusion proteins that account for up to 13–25% of PTCs[7,11]. Although BRAF mutations are prevalent in older patients, RET fusions are much more frequent in younger patients. RET fusions (also called *RET/PTC* rearrangements) are genomic rearrangements that are associated with ionizing radiation-induced DNA damage. RET fusions were reported in up to 60% cases of post Chernobyl PTCs[12]. Spatial contiguity of the genes involved in the fusion during interphase could be the structural basis of these chromosomal rearrangements[13]. In oncogenic RET rearrangements the kinase domain-containing C terminus of the RET gene, which is normally not expressed in thyroid follicular cells, is fused to the promoter-containing N terminus of a ubiquitously expressed, unrelated gene[14].

In this study, we aimed at the identification and characterization of the molecular events underlying PTC. By employing proteogenomic analysis of matching normal vs tumor vs lymph node metastasis of the same patient, we identified and validated a novel oncogenic RET fusion as well as other druggable targets in PTCs. We extended our proteomics observations by analyzing a cohort of PTC patient samples. Further, we provide mechanistic insights on the activation of the TFG-RET fusion and identified that E3 ubiquitin ligase HUWE1 is required for RET-mediated oncogenic transformation.

## Results

### Identification of a novel oncogenic RET fusion in a PTC patient.
From a cohort of PTC patients who are devoid of RAS and BRAFV600E mutations, a single patient who had a tumor mass largely in the right thyroid with multiple lateral lymph node metastases was selected. Tumor and LN metastatic tissue were harvested intraoperatively according to institutional guidelines with due ethical consent. Normal thyroid tissue from the left thyroid lobe was harvested during operation and served as a matching control. Histopathological analysis was performed to confirm the tumor content and the tissue specificity (Fig. 1a). In addition, α-calcitonin was detected in some cells of the primary tumor tissue implying c-cell hyperplasia and there was no α-calcitonin expression in the LN metastatic tissue (Fig. 1a). The tissue was lysed following a standard operating procedure as mentioned in the Methods section to collect the DNA, RNA and protein samples for subsequent genomics and proteomics analysis.

We next examined genomic alterations using both RNA-seq and exome-seq of the normal, tumor and LN metastasis samples. Exome data analysis of normal vs tumor and normal vs LN metastasis showed a total of 14 gene mutations of which 6 were shared between the tumor and LN metastasis (Supplementary Fig. 1A, B, Table 1). None of these mutations were well characterized mutations in known oncogenes or tumor suppressors. Recent studies suggest that more than 70% of PTCs harbor-activating mutations in BRAF, NRAS or HRAS[15]. We next examined RNA-seq data to look for other potential genomic drivers. RNA-seq-based fusion analysis detected a rearrangement where the 5′ end of the Trk fusion gene (TFG), which carries a PB1 domain, is fused to the 3′ kinase domain of the RET tyrosine kinase leading to a novel RET fusion (Fig. 1b). RET fusions involving other members have been characterized previously and occur in 7% of PTCs[15,16]. The fusion junction reads were detected only in the tumor and metastatic sample, thus confirming that the identified gene fusion is a somatic and not a germline event (Supplementary Fig. 1D, Supplementary Data 1). Differential expression analysis between the tumor and matched normal revealed 244 significantly deregulated genes (q-value < 0.001, Supplementary Data 2 and 3). Among these, we identified RET kinase as being significantly upregulated (Fig. 1c, Supplementary Fig. 1E). RET expression was at 14.7 and 10.2 RPMK in the LN metastatic and tumor sample, respectively, compared to 0.3 RPKM in the adjacent normal (Supplementary Fig. 1F). The average RET expression in normal thyroid tissue from GTEX (http://gtexportal.org) is 0.3 RPKM (Supplementary Fig. 1G). We examined previously described RET fusions in PTCs and found that RET fusions lead to significant overexpression of the RET gene (Supplementary Fig. 1H, p-value < 0.00001)[15]. Further, we isolated the mRNA from the patient's tumor lesion to generate cDNA, which was subjected to Sanger sequencing using primers covering the fusion region to confirm the presence of the fusion transcript in the tumor but not in the matching normal tissue (Supplementary Fig. 2A–C).

We then aimed to characterize the potential oncogenic activity of the RET fusion by stably expressing this gene fusion in immortalized human thyroid Nthy-ori 3-1 cells (from now on referred to as Nthy-TFG-RET cells). As expected, stable expression of TFG-RET fusion lead to increased viability (MTT assay) and cell proliferation (EdU assay) of Nthy-ori 3-1 cells (Fig. 2a, b and Supplementary Fig. 3A). Stable expression of TFG-RET cells lead to the transformation of these cells as revealed by soft agar colony formation assays (Fig. 2c). Further, TFG-RET expression activated several downstream pro-survival signaling pathways (Fig. 2d and Supplementary Fig. 3B). Our next step was to investigate whether TFG-RET expression could enable tumor formation in vivo. Subcutaneous injection of Nthy-TFG-RET cells into NOD/SCID mice, after a latency of about 12 weeks, showed tumor formation in mice carrying Nthy-TFG-RET cells, while the control group did not exhibit any tumor growth (Fig. 2e–g). Over the latency period, one shall not rule out the possibility that the Nthy-TFG-RET cells might have acquired additional mutations, which ultimately lead to tumor growth. However, the lack of tumor growth in the control group validates the oncogenic potential of TFG-RET. Together, these assays establish TFG-RET fusion as a thyroid oncogene.

### Characterization of TFG-RET.
In vitro kinase assays showed that the TFG-RET fusion exhibits constitutive kinase activity

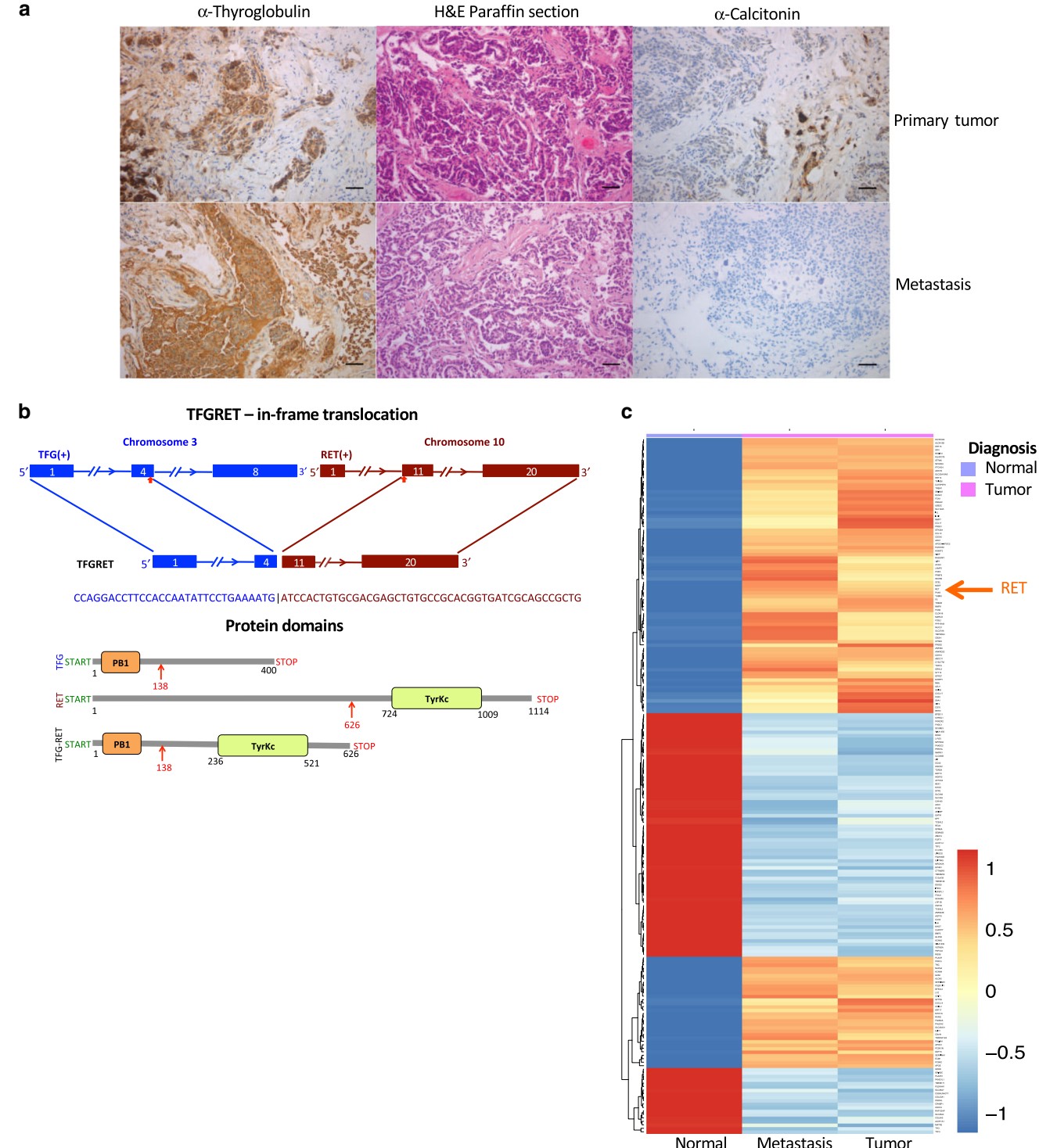

**Fig. 1 A novel fusion product is identified in patient PTC sample. a** Immunohistochemical analysis of patient samples. α-thyroglobulin expression was detected in both primary and LN metastasis tissues, implying that the tumor is a PTC. Haemotoxylin and eosin (H&E) staining shows follicular nature of the tumor. α-calcitonin was detected in some cells of the primary tumor tissue implying c-cell hyperplasia and there was no α-calcitonin expression in the metastatic tissue (magnification ×20, Bar 50 µm). Presented are representative data from a diagnostic staining procedure. **b** Diagrammatic representation of TFG-RET protein. TFG-RET contains 626 amino acids, and is a fusion between the N-terminal part of TFG protein and C terminus of RET protein. The TFG region (1–138) contains a PB1 domain, whereas the RET region (139–626) contains the tyrosine kinase domain. **c** Heat map of the RNA-seq analysis of normal vs tumor vs metastasis as mentioned in the Methods section. RET overexpression is indicated in the tumor.

(Fig. 3a). PB1 domains are evolutionarily conserved protein-protein interaction motifs contributing to formation of oligomers[17]. We hypothesized that the PB1 motif in the TFG-domain could possibly contribute to the formation of dimers and/or multimers of RET kinase, which is critical for the activation of the kinase. Immunoprecipitation of exogenously expressed RET constructs and the gene fusion revealed that TFG-RET readily forms dimers and multimers, which was confirmed by

**Table 1 Table listing the somatic mutations detected for the tumor and LN metastasis.**

| Gene_Name | Samples with mutation | (COSMIC_ID) | Total mutations in gene | Protein coding changes | Is a kinase? | Is a GPCR? |
|---|---|---|---|---|---|---|
| ADRA1D | Metastasis, tumor | | 1 | p.Arg282Cys | | Yes |
| MAP2K2 | Metastasis | | 1 | p.Gln391His | Yes | |
| ACCSL | Tumor | | 1 | p.Pro382Ser | | |
| ALCAM | Metastasis, tumor | | 3 | p.Arg304Lys(T/M),p.Arg304Ser(T/M), p.Ser324Pro(T/M) | | |
| ATP10A | Tumor | | 1 | p.Met1280Thr | | |
| MUC3A | Tumor | | 1 | p.Thr2064Asn | | |
| MYC | Tumor | | 1 | p.Ser21Asn | | |
| OR2L8 | Tumor | COSM226497 | 1 | p.Leu132Pro | | |
| OR5R1 | Metastasis, tumor | | 1 | p.Leu101Gln | | |
| PLEKHA5 | Metastasis, tumor | | 1 | p.Gln412fs | | |
| PPP1CC | Tumor | | 2 | p.Ile244Leu, p.Phe227Ser | | |
| RP1L1 | Tumor | COSM3762979 | 1 | p.Glu1343Lys | | |
| TG | Metastasis, tumor | | 1 | p.Thr1432_Ser1433del | | |
| ZFP42 | Metastasis, tumor | | 1 | p.Lys231Asn | | |

Enlists protein-altering gene mutations detected in exome analysis of tumor and LN metastatic lesions from the patients.

experiments employing crosslinking agents (Fig. 3b, c). As expected, the crosslinked wild-type kinase was not detected due to the limitation in the detection of larger oligomers with the employed gradient gels. Further, unlike the wild-type RET kinase, which is largely membrane-bound, a significant fraction of the TFG-RET fusion protein is detected in the cytosol of the cells stably expressing this gene fusion (Supplementary Fig. 3C, D). This could be attributed to the loss of membrane binding and other regulatory domains of full-length RET in the fusion, and is a characteristic of other RET fusions[14].

PB1 domains have been demonstrated to undergo head-to-tail dimerization[18]. DIX (dishevelled and axin) and PB1 domains are polymerizing domains that exhibit high structural similarities. We performed structural analysis of the TFG-PB1 domain by aligning it with the published structures of the DIX domain of Dvl2 and the PB1 domain of p62. We developed a TFG-PB1 model based on the DIX polymer of Dvl2, which displayed conserved residues and the coiled-coiled (CC) domain (Fig. 4a, b). We performed triple mutations (K14E, R22E and R23E) and deletion of the CC domain (Δ97-124) in TFG-RET. As expected, this lead to a reduced oligomerization of TFG-RET fusion as shown by gel filtration experiments (Fig. 4c–e). Further, mutations in the PB1 domain and in the CC domain led to reduced levels of phospho-RET (Y905), an autophosphorylation site of the RET kinase, which is also required for activity of RET kinase[19], as well as downregulation of other signaling pathways (Supplementary Fig. 4A–C). However, mutating three conserved residues in the PB1 domain but not in the CC domain compromised the stability of the protein (Supplementary Fig. 4D). We then performed soft agar colony formation experiments, which revealed that the integrity of the TFG-domain is not only required for TFG-mediated RET oligomerization but also for RET-mediated transformation (Fig. 4f). These data confirmed the critical role for TFG-domain-dependent oligomerization in RET-mediated tumorigenesis.

**Upregulation of ubiquitination-associated proteins in PTC.** With an aim to further identify factors that are differentially expressed in both tumor and LN metastatic lesions of the same patient, we performed label-free quantitative proteomics as described in the Methods section (Supplementary Data 4). These data uncovered several factors that are specifically expressed in tumors and LN metastatic lesions (Fig. 5a, Supplementary Fig. 5A

and Tables 2 and 3). We also observed that many of the factors that are upregulated on the mRNA levels are not detected on the protein levels, which suggest post-transcriptional regulation (Supplementary Data 5). Interestingly, we detected several members of the ubiquitin signaling machinery upregulated on the mRNA level in the tumor and LN metastatic lesion (Fig. 5 and Supplementary Fig. 5). We have extended the proteomic analysis further to 4 more patients and we consistently identified HUWE1 as one of the prime factors that are upregulated in the tumor and metastatic lesions (Fig. 5b, c, Supplementary Fig. 5C and Supplementary Data 6). We also detected STAT3 and KRAS being upregulated in the tumor and metastatic tissue of a subset of patients (Fig. 5c). As HUWE1 is more consistently detected in several patients, we focused further on the role of HUWE1 in mediating PTC tumorigenesis. HUWE1 is a HECT domain-containing E3 ubiquitin ligase that regulates the stability of various cellular targets and has been shown to exhibit both tumor suppressor and oncogenic functions[20,21]. For instance, HUWE1 can target oncoproteins like N-MYC, C-MYC, MCL1 and p53[22–25]. In addition, deubiquitinases like USP9X and UBP7 are also highly expressed in both tumor and LN metastatic lesions of the patient with TFG-RET fusion-expressing tumor tissue (Supplementary Fig. 5A, B). In order to test our proteomics observations in a larger cohort of patient samples, we analyzed the expression of HUWE1, USP9X and USP7 in protein lysates isolated from fresh frozen PTC patient samples (7 normal, 8 tumor, 3 metastatic lesions). Our data show that expression of HUWE1, USP9X and USP7 are higher in the tumor and metastatic lesions of many patients, compared to matched normal tissue irrespective of their mutational status (Supplementary Fig. 5D, E). We also observed an upregulation of HUWE1, USP9X and USP7 in Nthy-TFG-RET cells (Fig. 5d, Supplementary Fig. 5F). Expression of HUWE1 was enhanced on the mRNA level as well in Nthy-TFG-RET expressing cells (Fig. 5e).

We next investigated whether these ubiquitin-associated proteins have any functional role in cells expressing TFG-RET. For this, we employed a transient knockdown approach employing siRNAs against HUWE1, USP9X and USP7. We validated the knockdown efficiency of the siRNAs (2 sets against each target were used, with each set consisting of 2 different siRNAs; Supplementary Fig. 6A). Although HUWE1 and USP7 siRNAs were quite effective, USP9X siRNAs only displayed a 50% downregulation of the protein (Supplementary Fig. 6A). MTT

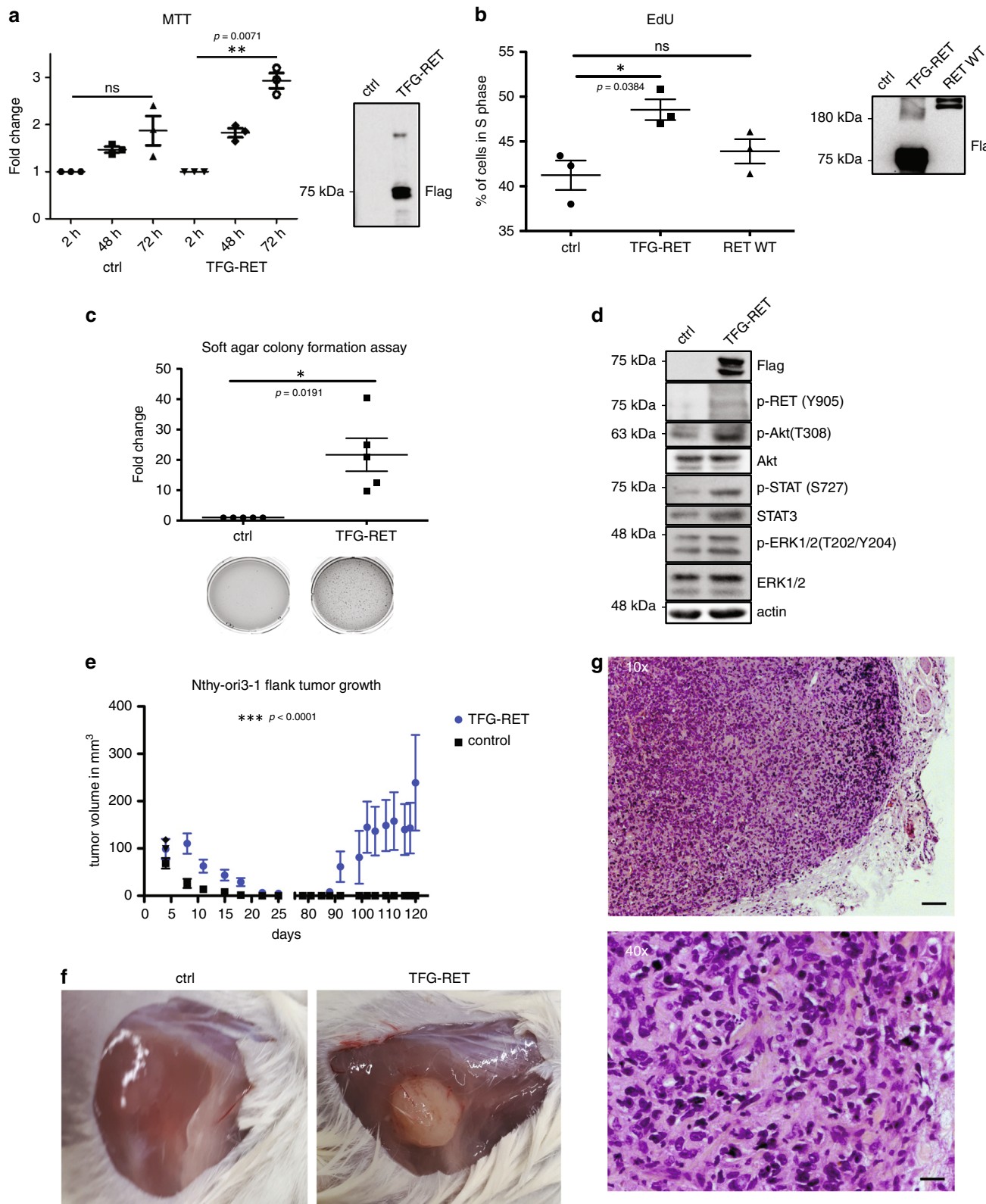

assays indicated that knockdown of HUWE1 and USP7 led to a significant reduction in the viability of Nthy-TFG-RET cells after 48 and 72 h (Fig. 6a and Supplementary Fig. 6B). In cellular proliferation experiments monitored by EdU incorporation and direct cell counting after 48 and 72 h, it was observed that HUWE1 knockdown significantly reduced cell proliferation,

although this was not always the case with USP9X and USP7 (Fig. 6a and Supplementary Fig. 6B, C). We hypothesized that HUWE1 and DUBs are probably required for RET-mediated transformation. Indeed, colony formation assays revealed a significant reduction in the transforming ability of Nthy-TFG-RET cells with HUWE1 knockdown (Fig. 6b). USP7 knockdown

**Fig. 2 TFG-RET expression induces increase in cell viability, cell proliferation and oncogenic transformation in vivo. a** Immortalized normal human primary thyroid follicular epithelial cells (Nthy-ori 3-1 cells) were infected with FLAG-tag expressing pPHAGE C-TAP TFG-RET virus particles and selected along with empty vector control. Cells were seeded in 96-well plates and subjected to MTT assay post 48 and 72 h as described in Methods. TFG-RET expression leads to an increase in cell viability after 72 h (absorbance relative to 0 h). Error bars represent ± SEM ($n = 3$). **$p < 0.01$, paired $t$-test, two-tailed distribution. Western blot confirmed TFG-RET expression. The fold increase in the absorbance (OD) was depicted as mentioned in the Methods section. **b** Nthy-TFG-RET cells along with empty vector control carrying cells were analyzed using EdU DNA synthesis assay as mentioned in the Methods section. Cells were seeded in 60 mm-cell culture plates and analyzed for cell proliferation after 72 h. Stable expression of TFG-RET leads to an increased cell proliferation. Error bars represent ± SEM ($n = 3$). *$p < 0.05$, paired $t$-test, two-tailed distribution. Stable expression of RET WT induced slight increase in cell proliferation, although this effect was not significant. The expression of wild type and fusion RET constructs were confirmed by western blots. Shown are the percentage of cells in the S phase for the analyzed samples. **c** Nthy-TFG-RET cells along with empty vector control were cultured in soft agar for 2 weeks followed by staining with crystal violet. Error bars represent ± SEM ($n = 3$). Paired $t$-test, two-tailed, three independent experiments: two with technical duplicates and one experiment with single technical replicate, $p$-value < 0.05. Shown are the fold changes in the number of colonies between the control and TFG-RET expressing cells. **d** Western blots for indicated proteins (p, phosphorylated) in lysates from Nthy-TFG-RET cells along with empty vector control. Expression of TFG-RET upregulated various cancer-associated signaling pathways. Shown are representative data from at least three independent experiments. **e** Growth of $4 \times 10^6$ subcutaneously injected thyroid Nthy-ori 3-1-cells in vivo. Nthy-TFG-RET cells formed solid, slowly progressing tumors in mice ($n = 8$ per group). For statistical analysis, two way ANOVA was performed. Error bars represent ± SEM. ***$p < 0.0001$. **f** Injection site of Nthy-ori 3-1-cells after 115 days of tumorigenesis. Nthy-TFG-RET injected mice developed solid tumors, whereas no tumors or neoplasia occurred in Nthy-ori 3-1 control injected mice ($n = 8$). **g** Histopathology of Nthy-TFG-RET tumors in H&E sections. The larger overview demonstrates distinct tumors with surrounding healthy tissue (scale bar = 100 μm). Higher magnification shows malignant growth patterns, atypical cell morphology and size, abrogation of the cell polarity and abnormal nuclei (scale bar = 50 μm). Shown here are representative images obtained from evaluation of H&E staining of multiple tumor tissue sections.

showed a tendency for reduced transformation, whereas USP9X did not show any such tendency at least in the time frame of the experiment (Supplementary Fig. 6C). This might be because of the lesser knockdown efficiency of the siRNAs employed against USP9X (Fig. 6a). We failed to obtain stable knockdown of these targets with shRNAs in Nthy-TFG-RET cells despite repeated attempts. Nevertheless, to further corroborate these observations, we employed small molecule inhibitors targeting HUWE1 (BI8622 and BI8626), DUBs (WP1130) and RET kinase activity (presented in Table 4). Treatment with these compounds significantly reduced RET-mediated colony formation (Fig. 6c). One of the suggested HUWE1 inhibitor (BI8622) almost completely inhibited the growth of TFG-RET transformed cells. We confirmed that both BI8622 and BI8626 inhibited HUWE1-mediated ubiquitination in cells by performing ubiqaptiure experiments (Supplementary Fig. 6D). Further, we also detected that these inhibitors reduced the proliferation of Nthy-TFG-RET cells, probably with a G1-S blockade, as they did not strongly induce cell death (Supplementary Fig. 7A–C). These results suggest that targeting the ubiquitin signaling machinery might possibly be explored as a strategy to combat RET-mediated oncogenesis in thyroid cancers.

## Discussion

The discovery of oncogenic gene fusions including BCR-ABL and ALK fusions has led to the development of successful targeted therapies, particularly in hematologic malignancies[26]. With the recent advances in next-generation sequencing techniques, there has been a massive increase in the number of molecular fusions described, especially in solid tumors. A TCGA study that aimed at the genomic characterization of 496 PTCs illustrated the mutual exclusivity of genetic driver alterations in PTCs, which further emphasizes the importance of precision medicine in the treatment of cancer[15]. PTC contributes to almost 80% of thyroid cancers and RET fusions are the most commonly detected gene fusions in PTCs[15]. More than 20 RET fusions have been reported in PTCs, of which the most common are the RET/PTC1 and RET/PTC3 fusions[27–29]. Recent studies have also identified RET fusions in lung and colon adenocarcinoma, breast cancer as well in MTCs, a more aggressive form of thyroid cancer[30,31]. By performing genomic analysis of a single PTC patient, we identified a novel, oncogenic RET fusion in both tumor and LN

metastatic lesions, whose expression readily transformed immortalized human thyroid cells and induced tumor formation in mice.

The PB1 domain in the TFG fragment has a key role in regulating the oligomerization and activation of the RET kinase. PB1 domains are capable of forming head-to-tail oligomers, which is a key step in signalosome assembly[18] and hence have an important role in the regulation of protein signaling. All DIX-like domains, including the PB1 domain have lower $K_d$ values, suggesting that they will produce oligomers in solution[18]. Mutation analysis showed that the PB1 domain-mediated oligomerization contributed to the autophosphorylation and/or activation of the RET kinase. Previous studies have shown that auto- and cross-phosphorylation at Tyr905 is required for RET kinase activity and mutation of this residue with others contributed to reduced RET activity[32]. Consistent with these observations, we detected that mutations in the oligomerization interface of the TFG-domain reduced Tyr905 phosphorylation in the TFG-RET fusion (Supplementary Fig. 4). Further, these mutations impaired RET-mediated oncogenic transformation (Fig. 4F). Together, this suggests that targeting the TFG-domain could possibly be a strategy to combat TFG-RET-mediated tumorigenesis, given the fact that most of the kinase inhibitors exhibit off target effects. Further guided-fusion detection studies in large cohorts of PTC patients are required to evaluate the frequency of this gene fusion. We also detected thus far unreported mutations in the coding regions of other interesting targets like the MAPK2K2 and several upregulated factors whose significance in driving PTC tumorigenesis needs further studies. Similar to the observation in the TCGA studies, we also detect that the frequency of the protein-altering gene mutations are in general low in PTCs (Supplementary Fig. 1C)[15].

By employing proteomics, we further demonstrated that several members of the ubiquitin signaling machinery are deregulated in both tumor and LN metastatic lesions and interestingly, we detected high HUWE1 expression in RET transformed cells. HUWE1 was also identified in patients without known RET fusions and further studies are clearly warranted (Fig. 5b). In addition to the kinome, the ubiquitinome has recently emerged as a favorable druggable target as there are several clinical trials going on with drugs targeting the ubiquitin signaling machinery[33]. Here, we present evidence that targeting HUWE1 or DUBs

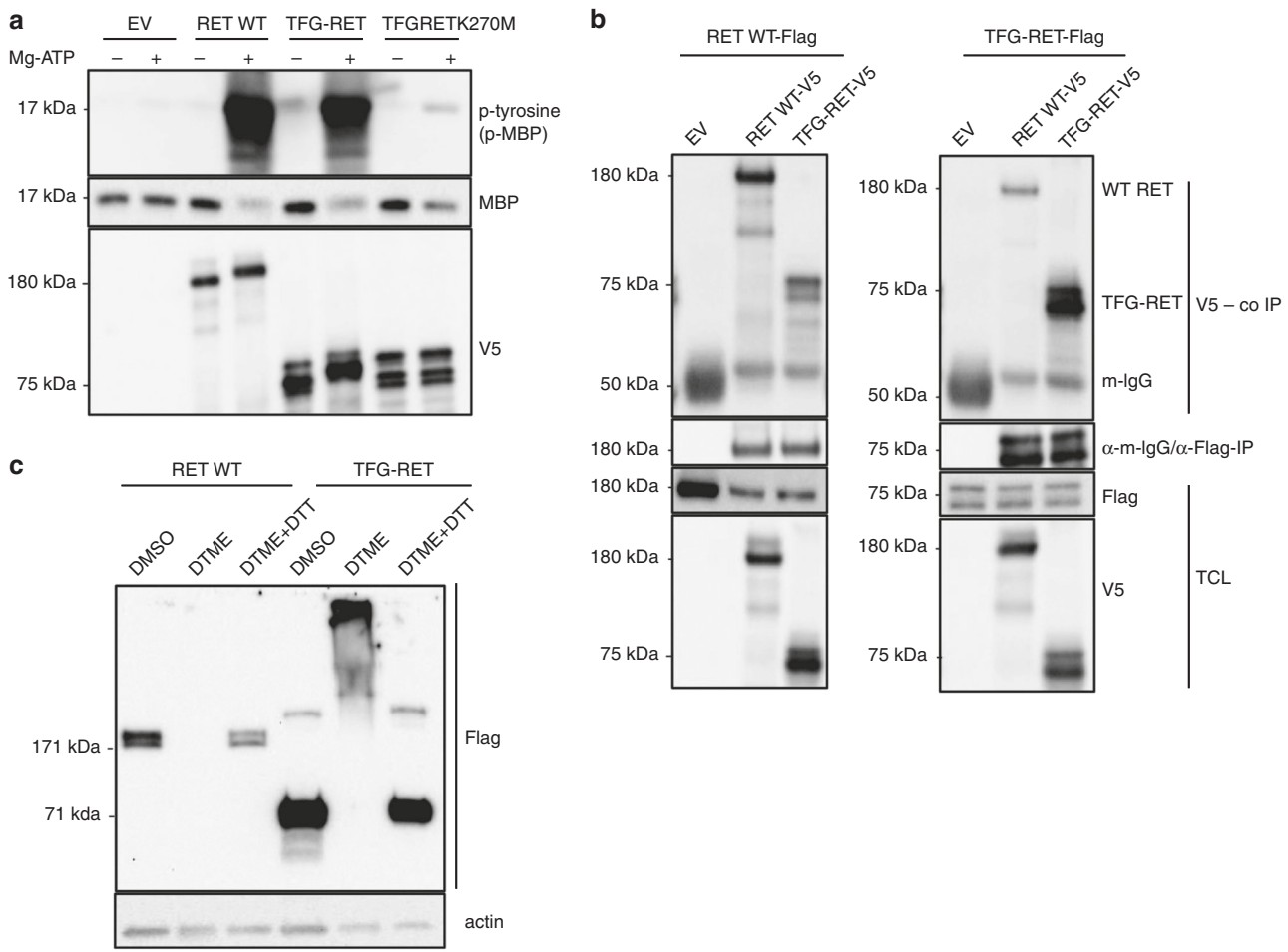

**Fig. 3 TFG-RET exhibits kinase activity and forms oligomeric complexes. a** HeLa cells were transiently transfected with V5-tagged wild-type RET, TFG-RET and kinase dead TFG-RET K270M along with empty vector. After 48 h cells were lysed and subjected to immunoprecipitation using anti-V5 antibody. After subsequent washes, the immunoprecipitated proteins were used for in vitro kinase assay, as described in Methods. TFG-RET phosphorylated MBP, thus exhibiting kinase activity like wild-type RET kinase. This kinase activity was abrogated in the TFG-RET kinase dead mutant (p, phosphorylated). Shown are representative data from at least three independent experiments. **b** HeLa cells were co-transfected with FLAG-tagged and V5-tagged wild-type RET and TFG-RET as indicated, followed by anti-FLAG immunoprecipitation after 48 h. Lysates were analyzed by western blotting (TCL, total cell lysate). Wild-type RET (FLAG®) co-immunoprecipitated with wild-type RET (V5) and also with TFG-RET (V5) (left panel), similarly, TFG-RET (FLAG®) also co-immunoprecipitated with both wild-type RET (V5) and TFG-RET (V5). Empty vector controls were included. Shown are representative data from at least three independent experiments. **c** Nthy-ori 3-1 cells stably expressing wild-type RET and TFG-RET were treated with DTME and DTME + DTT. Upon chemical crosslinking with DTME, similar to wild-type RET, TFG-RET also formed high molecular weight heteromeric complexes and these complexes were disrupted upon reduction with DTT (lane 3 and 6). Shown are representative data from at least two independent experiments.

like USP9X and UBP7 may be an attractive strategy to combat RET-mediated oncogenesis. The possible role of these ubiquitination-associated proteins in PTCs was further highlighted by our data from a larger cohort of PTC patient samples where these proteins seem to be overexpressed (Supplementary Fig. 5D, E). Whether these targets are also expressed in PTCs with other common mutations like BRAF, NRAS and NTRK1 fusions requires further analysis. It would also be very interesting to check whether upregulation of these proteins is correlative to RET fusions in general, not just limited to PTCs.

The determination of the malignancy status of thyroid nodules and accurate stratification of thyroid lesions are still challenging in the management of thyroid cancer. In a recent analysis of 56 patients suffering from PTC it was shown that RET rearrangements were associated with a higher risk of developing LN metastasis and an elevated risk of developing iodine refraction[34]. Improved molecular characterization of needle biopsies would ease stratification of thyroid nodules and would furthermore guide the development of "personalized therapeutics" for PTC

patients who are refractory to conventional RAI treatment modalities. Many of these targets described here could potentially be pursued for molecular diagnosis as well as patient stratification. Overall, our studies, in addition to unveiling an oncogenic gene fusion, have identified druggable targets, which open further avenues of treating PTCs, the most common type of thyroid malignancies.

## Methods

**Patient sample acquisition.** Informed consent was obtained from the patients prior to surgery. We have the ethical approval for the study granted by the "Landesärztekammer", which is also the institutional approval (Study number: 837.119.15 (9888)). Tumor and normal tissue were harvested intraoperatively from consented patients following standard operative procedures. For patient #1, the right thyroid lobe comprised mostly of tumor, while the left thyroid lobe predominantly was normal tissue. Ultrasound examination revealed that the patient had multiple lateral lymph node metastases. The staging was: pT4b (5.3 cm tumor), pN1b (17 metastases in 29 lymph nodes), M1 (pulmonary metastases), multifocal bilateral tumor, some with follicular pattern and also in the left lobe, capsular invasion but no invasion of adjacent structures. Routine Sanger sequencing was performed to monitor the mutational status of BRAF wild type and K, H, NRAS.

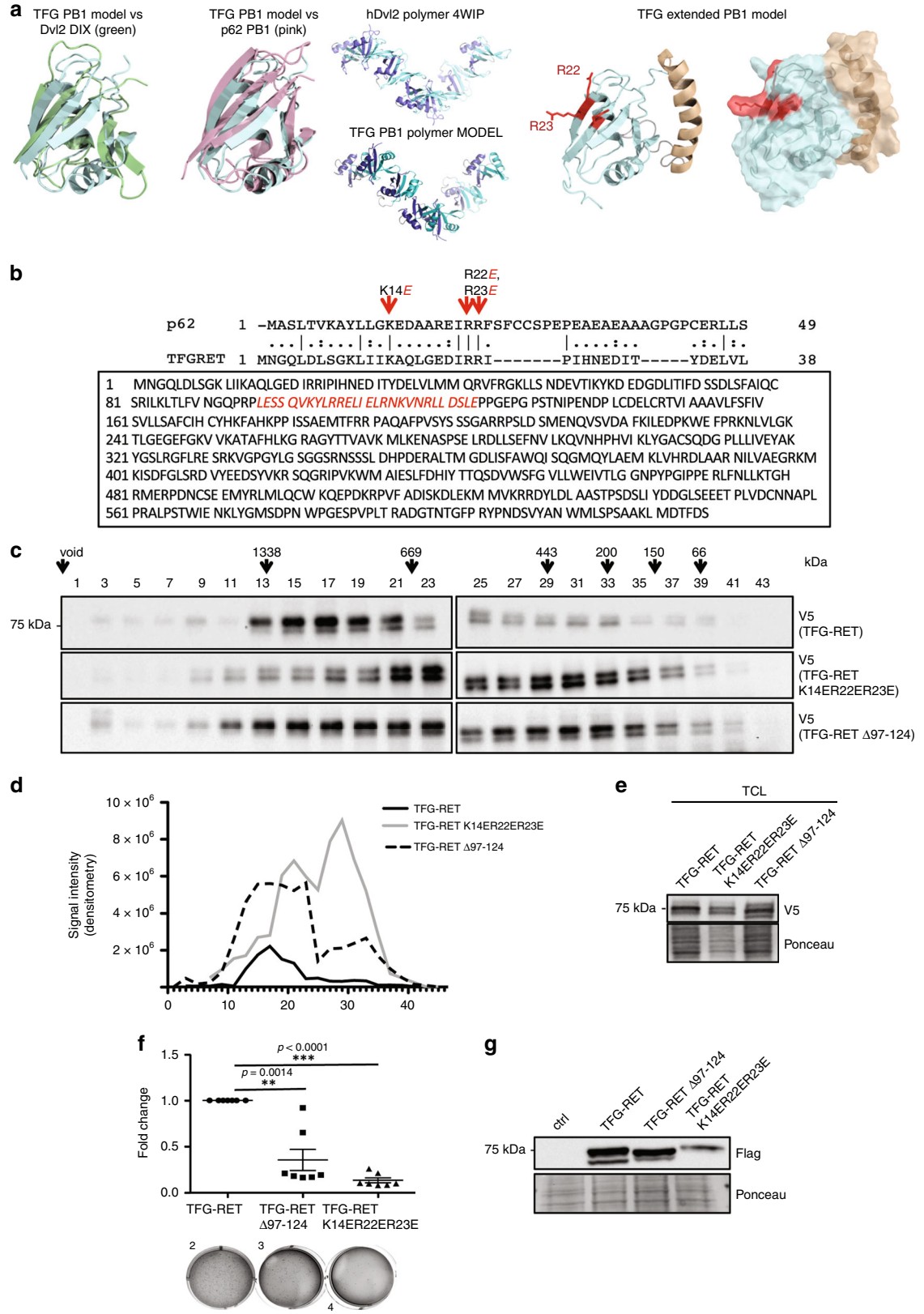

Normal, tumor and metastasic (from lymph node) tissue were taken for further analysis. A procedure involving a total thyroidectomy with therapeutic central and right lateral lymph node dissection (neck dissection, levels II–VII) was performed. The lymph node was chosen because the macroscopic appearance was clearly metastatic. Histology described a multifocal, bilateral follicular differentiated papillary thyroid carcinoma (pT4b, pN1, M1 (lung), Stage II, Classification according to 1997). All other tissue samples were provided by the tissue bank of the

University Medical Center Mainz in accordance with the regulations of the tissue bank.

**Isolation of protein, DNA and RNA from patient tissue**. The tumor tissue was incubated in collagenase-D solution (1 mg/mL) (11088866001, Roche) at 37 ℃ for 2 h and then transferred to a 6-well plate and cut into smaller pieces with a scalpel

**Fig. 4 TFG domains have a role in the TFG-RET heteromerization and oncogenicity. a** Structural comparison of TFG-PB1 vs Dvl2 DIX or p62 PB1 and the TFG-PB1 model showing the head region (deep blue) and tail region (teal) based on the DIX polymer of hDvl24WIP (head (slate), tail (aquamarine)). **b** Mutations in PB1 domain (top panel)—Comparison of PB1 sequences (not entire) of p62 and TFG-RET. Conserved residues Lys14, Arg22 and Arg23 were substituted with oppositely charged Glu to create PB1 domain mutant TFG-RET (TFG-RETK14ER22ER23E). CC domain (bottom panel)—Leu97 to Glu124 comprise the CC domain of TFG-RET. This domain was deleted to obtain the CC deletion mutant of TFG-RET (TFG-RET Δ97-124). **c, d** Lysates collected from HeLa cells transiently expressing indicated constructs were loaded on Superose-6 gel filtration column. Collected fractions (every second fraction) were subjected to western blotting. TFG-RET, TFG-RETK14ER22ER23E as well as TFG-RET Δ97-124 were detected using V5 tag. Most of TFG-RET was detected in high molecular weight fractions, while TFG-RETK14ER22ER23E shifted toward the lower molecular weight fractions. (Molecular weight corresponding to the elution volume are indicated by the arrows above). Shown are representative data from at least two independent experiments. **e** Protein expression of TFG-RET, TFG-RETK14ER22ER23E and TFG-RET Δ97-124 (pcDNA DEST V5.His vector) following transient transfection in HeLa cells was verified by Western blot analysis of the cell lysates (cytosolic fraction obtained after ultracentrifugation). Shown are representative data from at least two independent experiments. **f** Soft agar colony formation assay. Nthy-ori 3-1 cells stably expressing indicated constructs were cultured in soft agar for 2 weeks followed by staining with crystal violet. Error bars represent ± SEM ($n = 3$). Paired $t$-test, two-tailed. Three independent experiments were performed: one experiment with technical triplicates, and two experiments with technical duplicates. ***$p$-value < 0.0001, **$p$-value < 0.05. Shown are the fold changes in the number of colonies between the analyzed samples. **g** Immunoblot analysis of expression of TFG-RET, TFG-RET Δ97-124 and TFG-RETK14ER22ER23E in Nthy-ori 3-1 cells. Shown are representative western blots from the experiments presented in **f**.

and ground using the thumb press of a syringe plunger. The suspension was passed through a 70-μm cell sieve, centrifuged at $240 \times g$ for 5 min and subsequently washed with cold PBS. The cell pellet was re-suspended in lysing buffer (250 mM NaCl, 50 mM Tris-HCl pH 7.5, 10% glycerol, 1% Triton X-100, 0.2 mM $Na_3VO_4$ and protease inhibitor cocktail (539131, Calbiochem)) and kept on ice for 10 min. The cell suspension was sonicated (three times 10 s, 50% amplitude) and centrifuged at $17,000 \times g$, 4 °C for 10 min. Then, the supernatant was used for mass spectrometric analysis. DNA and RNA were isolated using AllPrep DNA/RNA Mini Kit (80204, Qiagen) following the manufacturer's protocol.

For western blot analysis of proteins from patients' tissue, samples—normal, tumor, metastasis—were lysed in RIPA buffer (25 mM Tris-Hcl (pH 8.0), 1 mM EDTA SDS, 1% Triton X-100, 0.1% Sodium deoxycholate, 0.1% SDS and 150 mM NaCl), sonicated and centrifuged. Supernatants were mixed with Laemmli buffer, boiled and subsequently analyzed by western blotting.

**Sequencing**. RNA-seq libraries were prepared using TruSeq RNA Sample Preparation v2 kit (RS-122-2001, Illumina). Exome capture was performed using the Agilent SureSelect Human All Exome kit (50 Mb; G3370L, Agilent). RNA-seq and Exome capture libraries were sequenced on HiSeq (Illumina) to generate $2 \times 150$ bp and $2 \times 75$ bp paired-end data, respectively.

**Exome data analysis**. Burrows-Wheeler Alignment (BWA) software set to default parameters was used to map sequencing short reads to UCSC human genome (GRCh37) using local realignment, duplicate removal and raw variant calling[35,36]. Strelka was used for somatic variant calling on tumor and its matched normal BAM file[37]. Known germline variants represented in the Exome Aggregation Consortium (ExAC) were filtered out[38].

**RNA-seq data analysis**. GSNAP (Genomic Short-read Nucleotide Alignment Program) was used to align RNA-seq reads to the human genome version NCBI GRCh37[39]. Differential gene expression analysis was performed using DESeq2[40]. Fusions were identified using a computational pipeline called GSTRUCT-fusions[41].

**cDNA isolation and PCR**. cDNA was synthesized from total RNA isolated from primary tumor of patient using RevertAid reverse transcriptase cDNA Synthesis Kit (EP0441, Thermo Fisher Scientific) according to the manufacturer's protocol. The cDNA was analyzed for the presence of TFG-RET fusion transcript by PCR amplification using primers that amplify the fusion region. PCR amplification of cDNA (diluted 1:10 in water) was carried out using Q5 High-Fidelity DNA polymerase (New England Biolabs, M0491L), in the presence of 10 mM DNTPs, 10 μM primers in 25 μL reactions. PCR reaction was set up as described below in a thermo cycler:

Denaturation: 98 °C—30 s
Amplification (35 cycles): 98 °C—10 s, 69 °C—30 s, 72 °C—15 s
Extension: 72 °C—2 min
Primer sequences:
TFG-RET 267 fwd–tgttaatggccagccaagac
TFG-RET 267 rev–taccacaagtttgcccacaa
TFG-RET 430 fwd–attagtgctaatgatgcaacgagt
TFG-RET 430 rev–cttgtgggcaaacttgttgg

The PCR products were subsequently subjected to gel electrophoresis in a 1% agarose gel and visualized under UV. For sequencing, the 430 bp PCR product was purified using QIAquick PCR purification kit (Qiagen, 28104) according to the manufacturer's protocol and sequenced using TFG-RET 430 fwd primer.

**Immunohistochemistry**. H&E staining was performed on formalin fixed paraffin embedded sections using standard laboratory procedures. Immunohistochemistry was performed on paraffin sections by using the DAKO-EnVision FLEX-kit (Dako, Glostrup, Denmark). Staining was performed on an immunostainer (Autostainer; Dako, Glostrup, Denmark) according to the manufacturer's instructions.

**Plasmids and constructs**. Wild-type RET and a kinase dead version of TFG-RET (TFG-RETK270M) were cloned into Entry vectors for subsequent gateway cloning into target vectors. pENTR221 TFG-RET, pENTR221 TFG-RETK14ER21ER22E and pENTR221 TFG-RETΔ97-124 were synthesized from Thermo Fisher Scientific and cloned into expression plasmids pPHAGE C-TAP (FLAG and HA tagged), a kind from Prof. Dr. Christian Behrends, and Gateway™ pcDNA™-DEST40 Vector (His and V5-tagged) (12274015, Thermo Fisher Scientific).

**Cell culture**. Nthy-ori 3-1 cells (90011609, Sigma) were cultured in RPMI-1640 medium supplemented with 10% heat inactivated FBS at 37 °C in 5% $CO_2$. HeLa (DSMZ) and 293T cells (a kind gift from Dr. Andreas Ernst) were cultured in DMEM supplemented with 10% heat inactivated FBS at 37 °C in 5% $CO_2$. For transient transfections, 5 μg plasmid and 27 μL of 10 mM polyethylenimine (PEI) were mixed in 500 μL of PBS and incubated for 15 min at room temperature. After incubation, transfection reagent was added dropwise to cells cultured in 100 mm plates. In order to generate Nthy-ori 3-1 cells empty vector/TFG-RET cell lines, we first transfected HEK293T cells with pPHAGE C-TAP, pPHAGE_CMV_C_-FLAG_HA_IRES_Puro together with the pLenti package (HDM-VSV-G; HDM-tatlb; HDM-Hgprn2 (gag-pol); RC-CMV-Rev1b) for lentiviral particle production. After 48 h, the media containing the virus were sterile filtered and then added to Nthy-ori 3-1 cells in the presence of 8 μg/mL polybrene. After 24 h, cells were selected with 2.5 μg/mL puromycin and the surviving pool of cells was expanded and maintained in puromycin (2.5 μg/mL) containing media.

**Quantitative RT-PCR**. Total RNA was extracted using TRIzol™ reagent (15596018, Thermo Fisher Scientific) according to the manufacturer's protocol. Equal amounts of total RNA were used to synthetize the corresponding cDNA using RevertAid reverse transcriptase cDNA Synthesis Kit (EP0441, Thermo Fisher Scientific). To quantify gene expression levels, SYBR-Green (A25780, Thermo Fisher Scientific) based qRT-PCR was performed using the StepOnePlus™ Real-Time PCR System (4376600). The expression level of HUWE1 normalized to expression of reference gene (18S or RSP13) was determined in triplicates.

**Primer sequences**. q-PCR primers:
HUWE1: Ref: NM_031407, for-TTGGACCGCTTCGATGGAATA, rev-TGAAGTTCAACACAGCCAAGAG
18s: Ref: NT_167214.1, for-AGAAACGGCTACCACATCCA, rev-CACCAGACTTGCCCTCCA
RPS13: Ref: NM_001017, for-CGAAAGCATCTTGAGAGGAACA, rev-TCGAGCCAAACGGTGAATC

**SDS–PAGE and western blot**. For SDS–PAGE, cell lysates with equal amounts of total proteins were prepared in SDS-sample buffer (0.125 M Tris-HCl, pH 6.8, 4% SDS, 10% glycerol, 10 mM DTT and bromophenol blue) followed by boiling at 95 °C for 10 min. Cell lysates were then loaded onto 4–15% mini-PROTEAN®TGX™ Precast Protein Gels (4561084, Bio-Rad) or 7.5% polyacrylamide gels. The proteins were then transferred to nitrocellulose blotting membranes (10600001, GE Healthcare) by western blotting. For immunoblot analysis, membranes were blocked with 5% low-fat milk in phosphate-buffered saline (PBS) for 30 min at

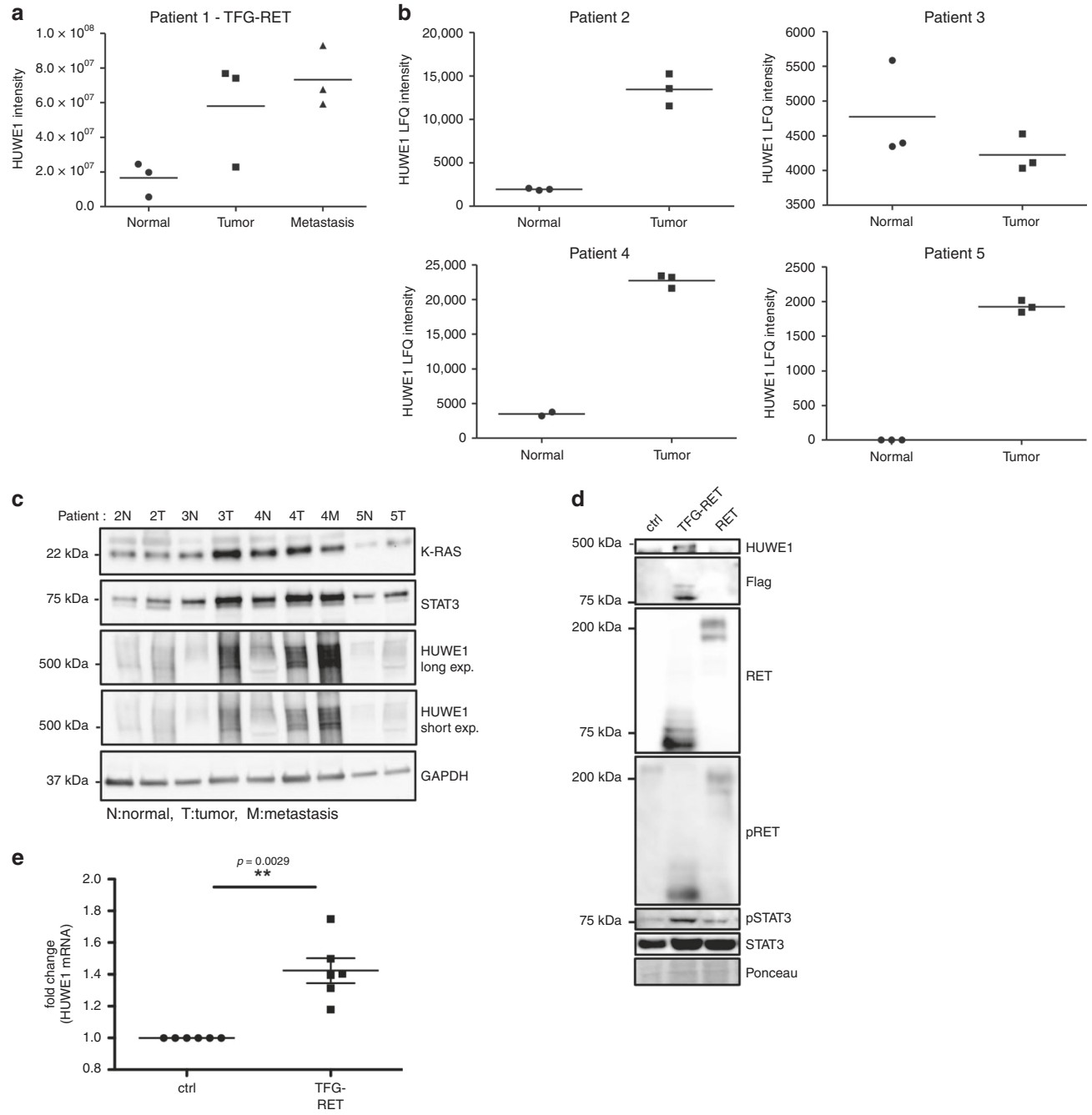

**Fig. 5 Ubiquitin pathway-associated proteins are upregulated in PTCs. a** Intensity values of HUWE1 from mass spectrometric data for patient 1. Shown are data from three technical runs from the same patient sample. **b** Intensity values of HUWE1 from mass spectrometric data for patients 2–5. Shown are data from three technical runs from the same patient sample. **c** Western blots showing expression of selected proteins (KRAS, STAT3, HUWE1) from patients 2–5. N = normal, T = tumor and M = metastasis. GAPDH was employed as a loading control. Shown are representative western blots from at least two independent experiments. **d** Western blots for indicated proteins (p, phosphorylated) in lysates from Nthy-ori 3-1 stably expressing FLAG-tagged (pPHAGE C-TAP vector) TFG-RET and RET. Cells transfected with empty vector served as control. Three independent experiments were performed (n = 3). **e** HUWE1 mRNA is overexpressed in Nthy-TFG-RET cells. Relative mRNA levels of HUWE1 were determined by real-time PCR. Error bars represent ± SEM (n = 3). Three independent experiments each with technical duplicates were performed. Paired t-test, two-tailed, p-value < 0.05.

room temperature and then incubated with primary antibodies in 3% bovine serum albumin (BSA; A7906, Sigma) overnight at 4 °C. Subsequently, the membranes were washed 3 times in PBS-T and incubated with horseradish peroxidase-coupled secondary antibodies for 1 h at room temperature, followed by washes as previously stated. The antigen–antibody complexes were detected by enhanced chemiluminescence (Immobilon Western Chemiluminescent HRP Substrate, WBKLS0500, Millipore) using Bio-Rad ChemiDoc™ Touch Imaging System (Bio-Rad). Quantification of Western blots was performed either by densitometry using the quantification software provided by Bio-Rad or by using Image J software (Open source image processing software http://imagej.net, Version: 2.0.0-rc-69/1.52i).

**Antibodies**. In this study, the following antibodies were used: Anti-thyroglobulin (M0781, Dako), anti-calcitonin (A0576, Dako) anti-phospho-tyrosine (P-Tyr-1000) MultiMab™ (8954S, CST), anti-Myelin Basic Protein (MBP) (13344, CST), anti-V5 antibody (Thermo Fisher Scientific, R960-25), anti-FLAG® M2-Peroxidase (A8592, Sigma), anti-phospho-RET (Y905) (3221S, CST), anti-RET (C31B4) (3223S, CST), anti-phospho-STAT3 (Ser727) (9134P, CST), anti-STAT3 (9139P, CST), anti-phospho-Akt (Thr308) (9275, CST), anti-Akt (C67E7) (4691, CST), anti-phospho-MEK1/2 (9154, CST), anti-MEK1 antibody (2352, CST), anti-phospho-p44/42 MAPK (Thr202/Tyr204) (ERK1/2) (9101L, CST), anti-p44/42 MAPK (ERK1/2) (9102, CST), anti-M2-PK (S-1, Schebo Biotech), anti-histone H3

**Table 2 Enlists the proteins identified only in tumor patient sample through mass spectrometric studies.**

| | Protein list—tumor only | | |
|---|---|---|---|
| | Accession | Entry | Description |
| 1 | P01258 | CALC_HUMAN | Calcitonin OS = Homo sapiens GN = CALCA PE = 1 SV = 2 |
| 2 | P22748 | CAH4_HUMAN | Carbonic anhydrase 4 OS = Homo sapiens GN = CA4 PE = 1 SV = 2 |
| 3 | P16671 | CD36_HUMAN | Platelet glycoprotein 4 OS = Homo sapiens GN = CD36 PE = 1 SV = 2 |
| 4 | P35908 | K22E_HUMAN | Keratin type II cytoskeletal 2 epidermal OS = Homo sapiens GN = KRT2 PE = 1 SV = 2 |
| 5 | Q16647 | PTGIS_HUMAN | Prostacyclin synthase OS = Homo sapiens GN = PTGIS PE = 1 SV = 1 |
| 6 | P63267 | ACTH_HUMAN | Actin gamma-enteric smooth muscle OS = Homo sapiens GN = ACTG2 PE = 1 SV = 1 |
| 7 | Q15326 | ZMY11_HUMAN | Zinc finger MYND domain-containing protein 11 OS = Homo sapiens GN = ZMYND11 PE = 1 SV = 2 |
| 8 | P55058 | PLTP_HUMAN | Phospholipid transfer protein OS = Homo sapiens GN = PLTP PE = 1 SV = 1 |
| 9 | P50440 | GATM_HUMAN | Glycine amidinotransferase mitochondrial OS = Homo sapiens GN = GATM PE = 1 SV = 1 |
| 10 | Q8IWL1 | SFPA2_HUMAN | Pulmonary surfactant-associated protein A2 OS = Homo sapiens GN = SFTPA2 PE = 1 SV = 1 |
| 11 | O00422 | SAP18_HUMAN | Histone deacetylase complex subunit SAP18 OS = Homo sapiens GN = SAP18 PE = 1 SV = 1 |
| 12 | Q86XX4 | FRAS1_HUMAN | Extracellular matrix protein FRAS1 OS = Homo sapiens GN = FRAS1 PE = 1 SV = 2 |
| 13 | P30566 | PUR8_HUMAN | Adenylosuccinate lyase OS = Homo sapiens GN = ADSL PE = 1 SV = 2 |
| 14 | P17661 | DESM_HUMAN | Desmin OS = Homo sapiens GN = DES PE = 1 SV = 3 |
| 15 | Q06828 | FMOD_HUMAN | Fibromodulin OS = Homo sapiens GN = FMOD PE = 1 SV = 2 |

**Table 3 Enlists the proteins identified only in LN metastatic patient sample through mass spectrometric studies.**

| | Protein list—metastasis only | | |
|---|---|---|---|
| | Accession | Entry | Description |
| 1 | Q04446 | GLGB_HUMAN | 1 4-alpha-glucan-branching enzyme OS = Homo sapiens GN = GBE1 PE = 1 SV = 3 |
| 2 | Q96C23 | GALM_HUMAN | Aldose 1-epimerase OS = Homo sapiens GN = GALM PE = 1 SV = 1 |
| 3 | Q96CN7 | ISOC1_HUMAN | Isochorismatase domain-containing protein 1 OS = Homo sapiens GN = ISOC1 PE = 1 SV = 3 |
| 4 | Q06124 | PTN11_HUMAN | Tyrosine-protein phosphatase non-receptor type 11 OS = Homo sapiens GN = PTPN11 PE = 1 SV = 2 |
| 5 | P32456 | GBP2_HUMAN | Guanylate-binding protein 2 OS = Homo sapiens GN = GBP2 PE = 1 SV = 3 |
| 6 | Q9NVE7 | PANK4_HUMAN | Pantothenate kinase 4 OS = Homo sapiens GN = PANK4 PE = 1 SV = 1 |
| 7 | Q53RD9 | FBLN7_HUMAN | Fibulin-7 OS = Homo sapiens GN = FBLN7 PE = 2 SV = 1 |
| 8 | Q9UDY2 | ZO2_HUMAN | Tight junction protein ZO-2 OS = Homo sapiens GN = TJP2 PE = 1 SV = 2 |
| 9 | P35237 | SPB6_HUMAN | Serpin B6 OS = Homo sapiens GN = SERPINB6 PE = 1 SV = 3 |
| 10 | P07358 | CO8B_HUMAN | Complement component C8 beta chain OS = Homo sapiens GN = C8B PE = 1 SV = 3 |
| 11 | Q6RW13 | ATRAP_HUMAN | Type-1 angiotensin II receptor-associated protein OS = Homo sapiens GN = AGTRAP PE = 1 SV = 1 |
| 12 | Q8WXX5 | DNJC9_HUMAN | DnaJ homolog subfamily C member 9 OS = Homo sapiens GN = DNAJC9 PE = 1 SV = 1 |
| 13 | Q16625 | OCLN_HUMAN | Occludin OS = Homo sapiens GN = OCLN PE = 1 SV = 1 |
| 14 | Q6P2P2 | ANM9_HUMAN | Putative protein arginine N-methyltransferase 9 OS = Homo sapiens GN = PRMT9 PE = 2 SV = 1 |
| 15 | P27361 | MK03_HUMAN | Mitogen-activated protein kinase 3 OS = Homo sapiens GN = MAPK3 PE = 1 SV = 4 |
| 16 | Q9UKG1 | DP13A_HUMAN | DCC-interacting protein 13-alpha OS = Homo sapiens GN = APPL1 PE = 1 SV = 1 |
| 17 | P17301 | ITA2_HUMAN | Integrin alpha-2 OS = Homo sapiens GN = ITGA2 PE = 1 SV = 1 |
| 18 | Q49A26 | GLYR1_HUMAN | Putative oxidoreductase GLYR1 OS = Homo sapiens GN = GLYR1 PE = 1 SV = 3 |
| 19 | Q9BXR6 | FHR5_HUMAN | Complement factor H-related protein 5 OS = Homo sapiens GN = CFHR5 PE = 1 SV = 1 |

(4499S, CST), anti-actin (ab49900, CST), anti-Na-K ATPase (MA3-928, Thermo Fisher Scientific), anti-HUWE1 (A300-486A, Bethyl), anti-USP7 (A300-033A, Bethyl), anti-USP9X (A301-351A, Bethyl) and anti-KRAS (c-166691, Santa Cruz).

**In vitro kinase assay**. $2 \times 10^6$ HeLa cells were seeded in 100 mm-cell culture plates and V5-tagged plasmids were transfected using PEI (as described in Cell culture and transient transfection) on the following day. 48 h post transfection, cells were lysed in lysis buffer (250 mM NaCl, 50 mM Tris-HCl pH 7.5, 10% glycerol, 1% Triton X-100 with protease inhibitor cocktail) and V5-tagged proteins were immunoprecipitated using V5 antibody and immobilized to agarose-coupled protein A/G beads (Roche, cat. nos. 11-134-515-001 and 11-243-233-001) overnight. Protein bound beads were washed with the lysis buffer and used for in vitro kinase assay. Kinase assay was performed using dephosphorylated myelin basic protein (MBP) (13-110, Merck) as a substrate in 25 mM Tris (pH 7.5), 5 mM β-glycerolphosphate, 2 mM DTT, 0.1 mM $Na_3VO_4$, 10 mM $MgCl_2$ and 5 mM ATP (Enzo, BML-EW9805-0100) in a final volume of 40 μL. The kinase assay mixture was incubated with the immunoprecipitated V5-tagged proteins for 30 min at 30 °C. The kinase reaction was terminated by adding 20 μL SDS-sample buffer (0.125 M Tris-HCl, pH 6.8, 4% SDS, 10% glycerol, 10 mM DTT and bromophenol blue) followed by boiling at 95 °C for 10 min. The samples were loaded onto 12% SDS–PAGE gels and subjected to immunoblotting analysis.

**Cell viability assay (MTT assay)**. Cell viability assay was performed by employing the Cell Proliferation Kit I (11465007001, Roche). Nthy-ori 3-1 cells stably expressing pPHAGE C-TAP empty vector and pPHAGE C-TAP TFG-RET were seeded in 96-well cell culture plates in 100 μL of complete growth medium (10,000 cells per well, triplicates per condition). Fold change was calculated as: Fold change = (O.D. at 48 h or 72 h-background)/(O.D. at 2 h-background). In case of siRNA-transfected cells, cells were harvested from 6-well cell culture plates 24 h post transfection, counted and then seeded in 96-well cell culture plates. Cellular viability was assessed after 2, 48 and 72 h by adding 10 μL of MTT solution for 2–4 h. After incubation, 100 μL of solubilization buffer was added to each well and incubated overnight in $CO_2$ incubator. Absorbance of the solubilized MTT was measured by absorbance plate reader (O.D. at 570 nm). For each condition, the average of three replicates was calculated. Then, the percentage of viable cells was determined as: Percentage of viable cells = 100*(OD of siRNA − OD of the background)/(OD of sicontrol − OD of the background).

**Soft agar colony formation assay**. 1.5% agarose solution was mixed with 2× growth medium (with 20% FCS, 2× inhibitor) to get a final mixture with 0.75% agarose in 1× growth medium (bottom agar medium). 1.5 mL of this bottom agar medium was added per well in a 6-well plate and incubated at room temperature for at least 10 min to solidify agarose. Nthy-ori 3-1 cells stably expressing pPHAGE C-TAP empty vector and pPHAGE C-TAP TFG-RET were diluted in 2× growth

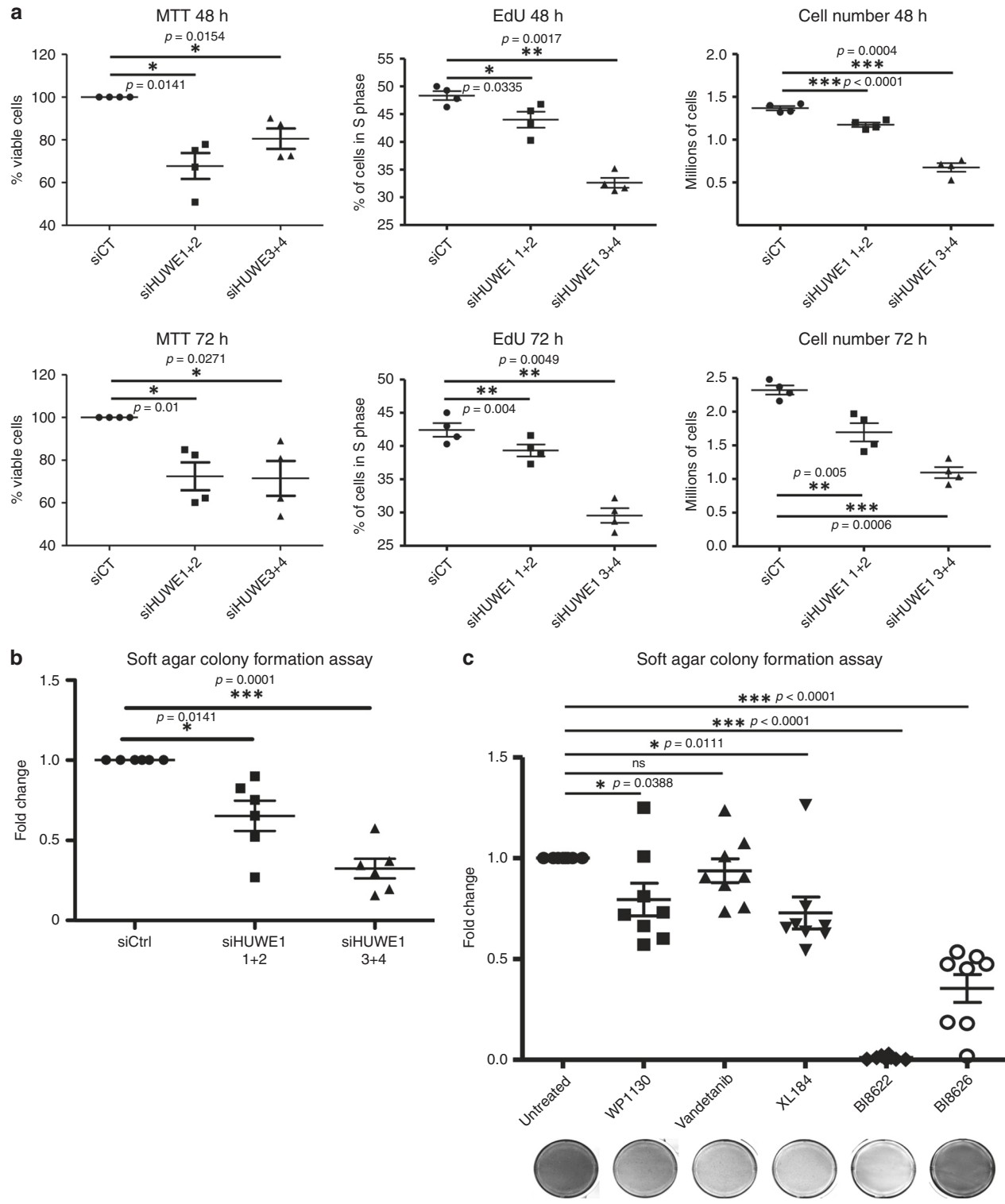

medium and mixed with 0.9 % agarose solution to a final concentration of 0.45% agarose. 1.5 mL of this cell suspension was added on the bottom agarose layer (5000 cells/condition). In case of siRNA-transfected cells, the cells were harvested 24 h post siRNA transfection from 6-well plates and then seeded for the soft agar assay. The cells seeded in soft agar were cultured for 2–4 weeks with addition of 100 μL of complete medium (with inhibitor) twice weekly. The colonies were stained with 0.02% crystal violet solution by gentle agitation at room temperature for about 30 min followed by washes with water. The images were taken with a ChemiDoc Touch (Bio-Rad) imaging system and the number of colonies was counted by Image J software. The anchorage independent growth of TFG-RET transformed cells was also tested and quantified in the presence of multiple inhibitors (WP1130 1 μM, Vandetanib 1 μM, XL184 2.5 μM, BI8622 20 μM, BI8626 20 μM).

**Dimerization experiments**. RET wild type, TFG-RET and TFG-RET mutant constructs in pPHAGE C-TAP (FLAG, HA tagged) and pcDNA3 Dest 40 (V5, His-tagged) were co-transfected as indicated in the results in HeLa cells using PEI (see Cell culture and transient transfection). 48 h post transfection, cells were lysed in lysis buffer (250 mM NaCl, 50 mM Tris-HCl pH 7.5, 10% glycerol, 1% Triton X-100 with protease inhibitor cocktail) and FLAG-tagged protein was immunoprecipitated using FLAG beads (Anti-FLAG® M2 Affinity Gel, A2220-5ML, Sigma). The co-precipitation of V5-tagged proteins was tested by immunoblots.

**Heteromerization studies**. Nthy-ori 3-1 cells stably expressing pPHAGE C-TAP RET wild type and pPHAGE C-TAP TFG-RET were seeded in 6-well cell culture

**Fig. 6 Inhibition of ubiquitination-associated proteins leads to reduced cell viability and transformation of TFG-RET expressing cells. a** HUWE1 was transiently knocked down using siRNA in Nthy-TFG-RET cells. The cells were seeded in 6-well and 96-well plates and subjected to EdU, cell counting and MTT assay after 48 and 72 h, respectively. Knockdown of HUWE1 resulted in a significant reduction of cell viability (MTT), proliferation (EdU) and cell number (cell counting). Error bars represent ± SEM ($n = 4$). Paired $t$-test, two-tailed, $p$-values: *<0.05, **<0.01 and ***<0.0001. For FACS analysis, the gating strategy is depicted in Supplementary Fig. 3A. For MTT assays, the percentages of viable cells are shown. For EdU assays, the percentages of cells in S phase are depicted. In the cell-counting experiments, the absolute viable cell numbers are shown. **b** HUWE1 was transiently knocked down using siRNA in Nthy-TFG-RET cells and cultured in soft agar for 2 weeks followed by staining with crystal violet. HUWE1 knockdown resulted in a significant reduction in the number of colonies. Error bars represent ± SEM. Three independent experiments with technical duplicates were performed ($n = 3$). Paired $t$-test, two-tailed, $p$-values: *<0.05, ***<0.0001. **c** Tyrosine kinase, DUB and HUWE1 inhibitors reduce oncogenic growth in TFG-RET expressing cells—soft agar colony formation assay. Nthy-ori 3-1 cells stably expressing TFG-RET were treated with inhibitors with indicated concentrations. Error bars represent ± SEM ($n = 3$). Paired $t$-test, two-tailed, $p$-value: *<0.05, **<0.01, ***<0.0001, ns not significant. In **b** and **c** the fold changes in the number of viable colonies from the analyzed samples are presented.

**Table 4 Summarizes the $IC_{50}$ values and effects/uses of the inhibitors used in the experiments.**

**List of inhibitors and targets/functions**

| Compound | $IC_{50}$ | Description |
|---|---|---|
| WP1130 | 1.25–5 μM (Usp9x) | Deubiquitinase inhibitor |
| XL184 (Cabozantinib) | 4 nM (RET) | RTK inhibitor, used in the treatment of MTC, RCC |
| Vandatenib | 40 nM (VEGFR2) 110 nM (VEGFR3) 500 nM (EGFR) | TK inhibitor, used in the treatment of MTC |
| BI8622 | 3.1 μM (HUWE1) | HUWE1 inhibitor |
| BI8626 2.5 μM | 2.5 μM (HUWE1) | HUWE1 inhibitor |

*RTK* receptor tyrosine kinase, *TK* tyrosine kinase, *MTC* medullary thyroid cancer, *RCC* renal cell carcinoma.

plates. Upon becoming 70% confluent, cells were treated either with DMSO (A3672.0250 Applichem), Dithio-bismaleimidoethane (DTME) alone (0.2 mM, 1 h; 22335, Thermo Fisher Scientific) or DTME followed by Dithiothreitol (DTT) (100 mM, 15 min). After thorough washes with PBS, cells were lysed in sample buffer (cells treated with only DTME were lysed in non-DTT containing sample buffer) and subjected to immunoblot analysis.

**Imaging studies.** Nthy-ori 3-1 cells stably expressing FLAG-tagged pPHAGE C-TAP TFG-RET or pPHAGE C-TAP RET were seeded on glass coverslips. The cells were transiently transfected with EGFP-C1 Lck-GFP (61099, Addgene) using PEI (as described in Cell culture and transient transfection). 48 h later, the cells were fixed using 4% formaldehyde for 10 min after media removal and two PBS washes. The cells were permeabilized using 0.1% Triton X-100 (3 min, room temperature). After two subsequent washes with PBS, the cells were blocked with 1% BSA for 30 min at room temperature. The cells were then stained for TFG-RET/RET using anti-FLAG® M2-Peroxidase (A8592, Sigma, 1:500 dilution in 1% BSA) for 1 h at room temperature. The cells were then washed with PBS and stained with anti-mouse Cy3 antibody (1:100 dilution in 1% BSA) along with Hoechst (2.5 μg/mL in 1% BSA) for 30 min in the dark at room temperature. The cells were washed with PBS and mounted on glass slides using Mowiol (+DABCO). Cells were imaged using a Leica SP8 confocal microscope (×63, oil immersion objective, Cy3 excitation at 552 nm, GFP excitation at 488 nm).

**Mice.** Female NOD.CB17-Prkscid mice were obtained from Janvier. All animals were housed at the animal facility of Johannes Gutenberg University using institutionally approved protocols (Landesuntersuchungsamt Koblenz). Animal procedures were performed under the supervision of the authorized investigators in accordance with the European Union normative for care and use of experimental animals.

**Subcutaneous tumor model.** Nthy-ori-3 cells or Nthy-TFG-RET ($4 \times 10^6$ cells in 200 μL PBS) were injected subcutaneously into the flank of 8- to 10-week-old female NOD.CB17-Prkscid mice. Tumor growth was observed over 115 days. Tumor sizes were determined by caliper measurements every other day. Tumor volume was calculated using the following formula: Volume = (width × length)/2. Exclusion criteria were tumor volumes exceeding 1 cm³, tumor sizes more than 1.5 cm in any direction and necrosis.

**Gel filtration.** HeLa cells were transiently transfected with TFG-RET, TFG-RET K14ER22ER23E and TFG-RET Δ97-124 in pcDNA3 Dest 40 (V5-, His-tagged). Approximately $1.5 \times 10^7$ cells were lysed 48 h post transfection in 500 μL lysis buffer (50 mM Tris pH 7.5, 150 mM NaCl, 10% glycerol, protease inhibitor cocktail (539131, Calbiochem)) by sonication (−50% amplitude, 5 s, four cycles). Cell

lysates were subjected to ultracentrifugation ($30,000 \times g$ for 1 h at 4 °C) and the supernatant (cytosolic fraction) was separated by size exclusion chromatography using an ÄKTA-Pure25 system equipped with two Superose-6 HR-10/30 columns (GE Healthcare, Freiburg, Germany) connected in series. The columns were pre-equilibrated with running buffer (50 mM Tris pH 7.5, 150 mM NaCl buffer, filtered 0.22 μm). 500 μL of cytosolic fraction were injected and separated at a flow rate of 0.5 mL/min. Fractions of 500 μL were collected and subjected to immunoblot analysis. Molecular weight reference proteins (Gel Filtration Marker Kit (mwgf1000, Sigma-Aldrich)) were separated under identical conditions and a calibration curve was obtained by plotting molecular weight (log scale) against elution volume.

**In-solution digestion and filter-aided sample preparation (FASP).** Samples for initial proteomic experiments were processed by in-solution digestion: Four volumes of ice cold acetone were added to the protein lysates (prepared as mentioned in the previous section (isolation of protein, DNA and RNA from patient tissue)), vortexed and precipitated at −20 °C overnight. Samples were centrifuged at $16,000 \times g$ for 20 min at 4 °C and the supernatant was discarded. Proteins were re-dissolved in 50 μL 6 M urea and 100 mM ammonium bicarbonate, pH 7.8. For reduction and alkylation of cysteines, 2.5 μL of 200 mM DTT in 100 mM Tris-HCl, pH 8 was added and the samples were incubated at 37°C for 1 h followed by addition of 7.5 μL 200 mM iodoacetamide for 1 h at room temperature in the dark. The alkylation reaction was quenched by adding 10 μL 200 mM DTT at 37 °C for 1 h. Subsequently, the proteins were digested with 10 μg trypsin (Promega) for 16 h at 37 °C. The digestion was stopped by adding 5 μL 50 % formic acid and the generated peptides were purified using OMIX C18, 10 μL (Agilent, Santa Clara), and dried using a Speed Vac concentrator (Concentrator Plus, Eppendorf).

The second proteomic dataset was processed using filter-aided sample preparation (FASP) as detailed before[42,43]. In brief, cells were dissolved in a buffer containing 7 M urea, 2 M thiourea, 5 mM DTT, 2% (w/v) CHAPS and lysed by sonication at 4 °C for 15 min using a Bioruptor (Diagenode, Liège, Belgium). The protein concentration was determined using the Pierce 660 nm protein assay (Thermo Fisher Scientific) according to the manufacturer's protocol. 20 μg of total protein were used for FASP. Proteins were transferred onto spin filter columns (Nanosep centrifugal devices with Omega membrane, 30 kDa MWCO; Pall, Port Washington, NY) and detergents were removed washing the samples three times with a buffer containing 8 M urea. After reduction and alkylation by DTT and iodoacetamide (IAA), excess IAA was quenched with DTT and the membrane was washed three times with 50 mM NH₄HCO₃. Afterwards, proteins were digested overnight at 37 °C with trypsin (Trypsin Gold, Promega, Madison, WI) using an enzyme-to-protein ratio of 1:50 (w/w). After digestion, peptides were recovered by centrifugation and two additional washes with 50 mM NH₄HCO₃. Combined flow-throughs were acidified with trifluoroacetic acid (TFA) to a final concentration of

1% (v/v) TFA and lyophilized. Purified peptides were reconstituted in 0.1% (v/v) formic acid (FA) for LC–MS analysis.

**Liquid chromatography–mass spectrometry (LC–MS)**. The tryptic peptides were dissolved in 10 μL 0.1% formic acid/2% acetonitrile and 5 μL were analyzed using an Ultimate 3000 RSLCnano-UHPLC system connected to a Q Exactive mass spectrometer (Thermo Fisher Scientific) equipped with a nano-electrospray ion source. For liquid chromatography separation, an Acclaim PepMap 100 column (C18, 2 μm beads, 100 Å, 75 μm inner diameter, 50 cm length) (Dionex, Sunnyvale CA, USA) was used. A flow rate of 300 nL/min was employed with a solvent B gradient of 4-35% in 180 min. Solvent A was 0.1% formic acid and solvent B was 0.1% formic acid/90% acetonitrile. The mass spectrometer was operated in the data-dependent mode to automatically switch between MS and MS/MS acquisition. Survey full scan MS spectra (from $m/z$ 400 to 2000) were acquired with the resolution $R = 70,000$ at $m/z$ 200, after accumulation to a target of 1e6. The maximum allowed ion accumulation times were 60 ms. The method used allowed sequential isolation of up to the ten most intense ions, depending on signal intensity (intensity threshold 1.7e4), for fragmentation using higher-energy collisional induced dissociation (HCD) at a target value of 1e5 charges, NCE 28, and a resolution $R = 17,500$. Target ions already selected for MS/MS were dynamically excluded for 30 s. The isolation window was $m/z = 2$ without offset. For accurate mass measurements, the lock mass option was enabled in MS mode.

For label-free quantification analysis, raw data were imported into PEAKS v8.5 (Bioinformatics Solutions Inc, Toronto, CA). Processed raw data were searched in PEAKS against the UniProt SwissProt database (Human, 20,251 proteins) assuming the digestion enzyme trypsin, at maximum two missed cleavage sites, parent ion tolerance of 10 ppm, fragment ion mass tolerance of 0.02 Da, carbamidomethylation of cysteines as fixed modification, and oxidation of methionines, deamidation of asparagine and glutamine residues as variable modifications. Label-free quantification was performed in the PEAKS software using a maximum mass difference of 15 ppm and a maximum retention time difference of 1.5 min for clustering and a 0.1% FDR threshold for peak annotation. For relative quantification, the data were normalized based on total ion current (TIC) in the PEAKS software to correct for unequal sample loading.

Proteomic samples (Patients #2, #3, #4, #5) prepared by FASP were analyzed by LC–MS on a Synapt G2-S HDMS mass spectrometer (Waters Corporation) coupled to a nanoAcquity UPLC system (Waters Corporation). Water containing 0.1% (v/v) FA, 3% (v/v) dimethyl sulfoxide (DMSO) served as mobile phase A and acetonitrile (ACN) containing 0.1% FA (v/v), 3% (v/v) DMSO as mobile phase B[44]. Tryptic peptides (corresponding to 200 ng) were loaded onto an HSS-T3 C18 1.8 μm, 75 μm × 250 mm reverse-phase column from Waters Corporation in direct injection mode. Peptides were separated at a flow rate of 300 nL/min applying a gradient from 5 to 40% (v/v) mobile phase B over 90 min. Afterwards, the column was washed with 90% mobile phase B and re-equilibrated to initial conditions resulting in a total analysis time of 120 min. The column was heated to 55 °C. Eluting peptides were analyzed in positive mode ESI-MS by ion-mobility separation (IMS) enhanced data-independent acquisition (DIA) UDMS$^E$ mode as described before[43,45]. Acquired MS data were post-acquisition lock mass corrected using [Glu1]-Fibrinopeptide B, which was sampled every 30 s into the mass spectrometer via the reference sprayer of the NanoLockSpray source at a concentration of 250 fmol/μL.

LC–MS DIA raw data were processed and searched with ProteinLynx Global SERVER (PLGS) (version 3.02 build 5, Waters Corporation) against a custom compiled database containing UniProtKB/SwissProt entries of the human reference proteomes (entries: 20,394) as well as common contaminants. Following search criteria were applied: (i) Trypsin as digestion enzyme allowing up to two missed cleavages, (ii) carbamidomethyl cysteine was defined as fixed and (iii) methionine oxidation as variable modification. The false discovery rate (FDR) for peptide and protein identification was assessed searching a reversed database and set to a 1% threshold for database search in PLGS. Label-free quantification analysis was performed using ISOQuant as described before[45]. For each protein, absolute in-sample amounts were estimated using TOP3 quantification[46]. The mass spectrometry proteomics data have been deposited to the ProteomeXchange consortium PRIDE under two data set identifiers (a) PXD016828 (patient #1) and (b) PXD016739 (patients #2-#5).

**Endogenous ubiquitination experiments**. Nthy-ori 3-1 cells stably expressing pPHAGE C-TAP TFG-RET (in 100 mm-cell culture dishes, about 70% confluent) were treated either with DMSO, 20 μM BI8622 or BI8626 (synthesized by Syngene International limited, India) inhibitors for 2 h followed by treatment with 10 μM MG132 for 5 h at 37 °C. Following treatment, cells were lysed in lysis buffer and 250 μg protein was used to isolate ubiquitinated protein using UBIQAPTURE-Q® kit (BML-UW8995-0001, Enzo) according to the manufacturer's protocol. Ubiquitination of proteins was determined by immunoblot analysis.

**Cellular fractionation assay**. Nthy-ori 3-1 cells stably expressing pPHAGE C-TAP RET wild type and pPHAGE C-TAP TFG-RET were cultured in 100 mm dishes. 48 h post seeding, the growth media were removed from the culture dish and the cells were washed with cold PBS. Cells were subsequently lysed using

buffers from ProteoExtract® Subcellular Proteome Extraction Kit (539790, Merck) according to the manufacturer's protocol. Lysates collected were subjected to immunoblot analysis.

**Cycloheximide-chase assay**. Nthy-ori 3-1 cells were transiently transfected using PEI (as described earlier) with different plasmids in 100 mm dishes. 24 h post transfection, cells were re-seeded in 12-well plates. After 24 h, cycloheximide (100 μg/mL) was added to the cells and samples were collected in SDS-sample buffer at indicated time points and boiled at 95 °C for 10 min. Cell lysates were subjected to immunoblot analysis.

**EdU DNA synthesis assay**. Nthy-ori 3-1 cells stably expressing pPHAGE C-TAP empty vector, pPHAGE C-TAP TFG-RET or pPHAGE C-TAP RET were seeded in 60 mm-cell culture plates (0.4 × 10$^6$ cells per plate). EdU DNA synthesis assay was performed using Click-iT® Plus EdU Pacific Blue™ Flow Cytometry Assay Kit (Thermo Fisher Scientific, C10636) according to the manufacturer's protocol. Briefly, 72 h post seeding, the cells were incubated for 2 h in 10 μM EdU (5-ethynyl-2′-deoxyuridine). The cells were then harvested, fixed, permeabilized and stained with pacific blue ligand (for incorporated EdU) and propidium iodide. Subsequently, pacific blue-positive cells were measured using a flow cytometer (405 nm laser). For EdU assays in cells after transient siRNA knock down, cells were seeded in 60 mm-cell culture plates and transfected with siRNA on the next day (as described in siRNA transfection). Total cell number was determined using an automated cell counter (Bio-Rad TC20™ Automated Cell Counter, 1450102) to normalize the starting cell numbers. EdU DNA synthesis assay was performed 48 h and 72 h post transfection.

**siRNA transfection**. ON-TARGET plus Human siRNAs (set of 4 per target) were purchased from Dharmacon (details below). Two sets of siRNAs (each set with 2 different siRNAs) were used for each protein target. Cells were transiently transfected with siRNA using SAINT-sRNA transfection reagent (SR-2003-01, Synvolux Products) according to the manufacturer's protocol. Briefly, cells were seeded at a density of 0.2 × 10$^6$ cells per well of a 6-well plate. On the next day, 200 μL PBS, 1.25 μL of each of the two siRNAs (20 μM) and 20 μL of SAINT-sRNA transfection reagent were mixed and incubated in room temperature for 15 min to form complexes. After changing the medium to serum-free RPMI-1640 medium, the siRNA-SAINT-sRNA complexes were added dropwise to the cells. The medium was replaced with fresh RPMI-1640 containing 10% FBS. The samples were collected for Western blot analysis 48 h post transfection.

ON-TARGETplus non-targeting siRNA
siRNA D-001810-01-05 UGGUUUACAUGUCGACUAA
siRNA D-001810-01-06 UGGUUUACAUGUUGUGUGUGA
ON-TARGETplus Human HUWE1 siRNA LQ-007185-00-0005
J-007185-07, HUWE1 GCUUUGGGCUGGCCUAAUA
J-007185-08, HUWE1 GCAGUUGGCGGCUUUCUUA
J-007185-09, HUWE1 GAGCCCAGAUGACUAAGUA
J-007185-10, HUWE1 UAACAUCAAUUGUCCACUU
ON-TARGETplus Human USP7 siRNA LQ-006097-00-0005
J-006097-05, USP7 AAGCGUCCCUUUAGCAUUA
J-006097-06, USP7 GCAUAGUGAUAAACCUGUA
J-006097-07, USP7 UAAGGACCCUGCAAAUUAU
J-006097-08, USP7 GUAAAGAAGUAGACUAUCG
ON-TARGETplus Human USP9X siRNA LQ-006099-00-0005
J-006099-06, USP9X AGAAAUCGCUGGUAUAAAU
J-006099-07, USP9X ACACGAUGCUUUAGAAUUU
J-006099-08, USP9X GUACGACGAUGUAUUCUCA
J-006099-09, USP9X GAAAUAACUUCCUACCGA

**Statistical analysis**. Data were statistically analyzed by using GraphPad Prism software (Prism 5 for Mac OSX, version 5.0a). Tests employed are mentioned in the figure legends.

**Reporting summary**. Further information on research design is available in the Nature Research Reporting Summary linked to this article.

## Data availability

The data are available from authors upon request. The proteomics data are available through PRIDE (under accession numbers PXD016739 and PXD016828). Sequencing data were submitted to the EBI ENA database under accession id PRJEB37220 (ERP120526).

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

## Acknowledgements

We thank all the patients who have volunteered and donated their biomaterials for the study. We thank Stefanie Wenzel for excellent technical assistance. We thank Dr. Stefanie Zimmer and the biobanking team for their help with the consenting and collection of patient materials. We thank Prof. Dr. Christian Behrends for the pPHAGE vector and Dr. Andreas Ernst for 293T cells. We thank Dr. Christiane Schoenfeld for assisting with the animal experiments and for critical reading of the manuscript. This work is primarily supported from a DKH grant (DKH 70113042) to K.R. and from the CRC 1292 (TP12) and GFK fellowship to K.R. K.R. is supported through a Heisenberg professorship of the DFG (RA1739/4-1). K.R. also acknowledges support from FZI (Forschungszentrum Immuntherapie) of the UMC-Mainz.

## Author contributions

A.K., J.B. and E.R. designed and performed most of the experiments, analyzed/interpreted data and prepared figures. S.S., Z.M., J.W. and E.S. contributed to genomic studies. U.D., S.T. and B.T. performed proteomics mass spectrometric analysis. M.F. and M.B. contributed to the PB1 modeling studies, data analysis and manuscript preparation. S.R. performed animal experiments. A.S. and W.R. performed and contributed to the analysis of tissue pathology. T.J.M. performed surgery, contributed to study conception-design and analysis. K.R. contributed to the conception, design, analyzed/interpreted data and supervised the study. K.R., A.K., J.B. and E.R. wrote the manuscript with input from all authors.

## Competing interests

The authors declare no competing interests.
