## [Peer Review File · Nature Communications]

Reviewers' comments:

Reviewer #1 (Expertise: Papillary thyroid carcinoma, Remarks to the Author):

The Authors compared DNA, RNA, and protein expression among adjacent (normal) thyroid, primary papillary thyroid cancer (PTC), and its metastatic lymph node (LN met) from one patient and found that a novel RET fusion gene, TFG-RET, in primary lesion and LN met. They examined the function of TFG-RET and demonstrated that TFG-RET transfected NThyOri3 cells were transformed and fusion caused dimerization of the protein, resulting activation of downstream signaling. They also found that involvement of ubiquitin machinery was dysregulated by TFG-RET and its inhibition abolished transformation induced by TFG-RET. Therefore, the Authors concluded that ubiquitin signaling machinery may be potential therapeutic target for PTC.

This is important discovery and interesting observation. However, there were many concerns as below.

[GENERAL CONCERNS]

1. Discovery of a new fusion gene, TFG-RET, was not confirmed by sequencing and its functional role in PTC was not clear.
2. "metastasis" was used for LN metastasis. Since metastasis includes LN and distant metastasis, it was better to use LN metastasis to distinguish from distant metastasis.
3. Some description was too short to explain data, including Results, Materials and Methods, and Figure legends, and it was difficult to follow
4. Use the word consistently. For example, tumour and tumor, and 100 mm and 10 cm, were used in this manuscript.
5. Company information was not completed in many section in Materials and Methods.

[SPECIFIC CONCERNS]

6. Introduction: Between Line 61 and Line 80 was too long to explain other oncogene in PTC. Instead, it is better to review RET fusion protein more.
7. Line 96: What is SOP? Standard operation procedure? Please describe specific procedure.
8. Line 100: Exome data give us gene information, not protein. Please use the word "14 gene" instead of "14 protein". This applies to Figures and Figure legends.

9. Figures S1A and S1B, and Table 1: The Reviewer did not understand figure S1B and Table 1. Tumor had 16 and metastasis had 9 fusion gene? How these genes were listed in Table 1? Table 1 showed 13 in tumor and 7 in LN. Please explain more clearly either in Results or legends.
10. TFG-RET fusion was identified by RNA-seq analysis. Did the Authors confirmed this fusion gene by sequencing of gene from primary cancer and LN metastasis?
11. Supplementary table 1: Table showed 1734 genes. It was better to make another table including only 244 genes.
12. Figure C: Figure lacks a labelling of primary and LN. It was not sure whether RET overexpression was due to TFG-RET. Did the authors examine other RET fusion genes and RET gene?
13. Line 113: Stably transfected NThyOri3 cells were pooled cells or single cell-derived clone? Please clarify them. If single clone was used, at least one more clone should be tested.
14. Figure 2A: Since MTT assay is cell viability assay, not cell proliferation assay, data from different day should not compared. It is possible to be affected by cell density, medium pH, and other factors. Please confirm this data by counting cell number.
15. Figures 3C and 3D: Figure legend for 3C was wrong. Please explain the number at the top of figure. If the molecular size TFG-RET protein were 75 kDa, why were they eluted much earlier? Please explain this.
16. Figure S3A: It was important to show some downstream protein changes, such as pERK and pAkt in addition to pRet to support functionally activating.
17. Line 141: Fig. S2E must be Fig S3C?
18. Line 153: Please change the word, "deregulated" to more specific word, "increased" or "decreased".
19. Line 172: In thyroid cancers, many fusion proteins, including RET, BRAF, ALK, NTRK, PPARgamma, were reported and characterized. This sentence made confusion, and gave different message.
20. Line 179: Reference 26 listed only 12 RET fusion proteins. Please cite right reference, which showed more than 20 RET fusion proteins.
21. Patient: Consent was described, but institutional approval was not clearly described. What was the surgical procedure? Total thyroidectomy and neck LN dissection? Please specify the surgery.
22. Line 229: The Authors obtained only one LN? Why this one LN was chosen?. Did they perform confirmatory experiments using DNA from other LNs?
23. Line 258: There was other section describing antibodies. Please move to there or change the description. Ex. Antibodies against thyroglobulin and calcitonin were purchased from DAKO.....
24. Invitrogen and Thermo Fisher are the same company in the US. Please use Thermo Fisher.
25. Line 271: Please explain PEI.

26. Line 273: What were experiments carried out 48 hours post transfection. From this sentence, it was not clear.
27. Line 284: How were gene expression normalized? Please explain.
28. Line 298: Missing the information of antibody A7906 from Sigma.
29. Line 328, Cell proliferation assay: MTT is not the proliferation assay. This is cell viability assay.
30. Line 344: Replace to with on.
31. Line 348: Please add the source of Image J.

Reviewer #2 (Expertise: DUBs, Proteomics, Remarks to the Author):

The authors present the analysis of a single thyroid tumour of a patient with TFG-RET fusion. They show that the TFG-RET oncogene oligomerises and this is important for transformation. Proteomics analysis of the tumour shows up-regulation of ubiquitin ligase HUWE1 as well as DUBs Usp9X and UBP7. Over-expression of TFG-RET in cell culture leads to up-regulation of HUWE1 and pharmacological inhibition of RET. HUEW1 and DUBs reduced oncogenesis.

Major points.

- The authors' main problem is that they have n=1. Proteomics and transcriptomics analyses with one bioreplicate are rather problematic.
- Figures are not well described. It would be much better if the figure panels included small titles such as "MTT assay" or "proliferation" (Figure 2).
- For the proliferation assays, the over-expression of the WT RET is missing. It is quite obvious that over-expressing an oncogene leads to more proliferation, also when the oncogene is fused to TFG.
- Some of the data is just not good enough to draw the conclusions that the authors have drawn. For example, Figure S4B appears to present three replicates of cell cultured cells overexpressing TFG-RET. The loading control vinculin is off, USP7 is all over the place and none of the proteins HUWE1, USP7 and USP9X really changes reliably.
- The paper is rather too concisely written and thus, one seems to miss details. The methods are too short for anyone to reproduce. Are the replicates in proteomics technical replicates and at which level? Reinjection of the same sample? Details of the inhibitor treatment is missing in methods.

- The volcano plot in figure indicates non-equal loading of samples as everything is up in the tumour. How did the authors normalise?
- The DUB and HUWE1 inhibitors are unspecific. For WP1130 you can look here: PMID: 25159004. There is a published USP7 inhibitor that is specific and you can find the paper here: PMID: 29045385. It is not surprising that these unspecific inhibitors kill the cells. You would need to compare treated cells overexpressing RET-TFG versus plasmid control and overexpressing RET alone.
- shRNA experiments of these enzymes will be more reliable.

Reviewer #3 (Expertise: Gene fusions, RNA sequencing, Remarks to the Author):

Krishnan et al performed exome and transcriptome sequencing on a triplet of a thyroid tumor, a metastatic lesion, and an adjacent normal sample. A RET fusion was identified that has not been previously described, and using several functional approaches this fusion was found to transform immortalized cells. Small molecule inhibitors of RET and downstream proteins were shown to reduce RET-mediated oncogenesis. RET fusions have recently gathered more interest due the favorable responses seen in patients with RET-fusion positive tumors after treatment with for example LOXO-292.

Addressing the following questions would help to provide context to this work.

1. TFG fusions are often caused by a germline copy number variant. Can the fusion be validated genomically, and be shown to be unique to the tumor?
2. Resources such as tumorfusions.org and ChimerDB have enumerated fusions across many cancers. How often is the same TFG-RET fusion seen in other tumors?
3. What is the expression of the fusion transcript relative to full RET transcript.
4. Fig. 1C. Legend refers to exome analysis as well as overexpression. What does this figure represent, exome sequencing does not measure gene transcription.

We thank the reviewers for their time, constructive comments and suggestions on our work. We take the comments very positively and have addressed the concerns with the best possible efforts. We believe that this has made the manuscript clearer and stronger.

Please find below our responses to the reviewer's comments.

Reviewer #1 (Expertise: Papillary thyroid carcinoma, Remarks to the Author):

The Authors compared DNA, RNA, and protein expression among adjacent (normal) thyroid, primary papillary thyroid cancer (PTC), and its metastatic lymph node (LN met) from one patient and found that a novel RET fusion gene, TFG-RET, in primary lesion and LN met. They examined the function of TFG-RET and demonstrated that TFG-RET transfected NThyOri3 cells were transformed and fusion caused dimerization of the protein, resulting activation of downstream signaling. They also found that involvement of ubiquitin machinery was dysregulated by TFG-RET and its inhibition abolished transformation induced by TFG-RET. Therefore, the Authors concluded that ubiquitin signaling machinery may be potential therapeutic target for PTC.

This is important discovery and interesting observation. However, there were many concerns as below.

We thank the reviewer for highlighting the importance and significance of our observation. We also highly appreciate the caution to details, which has helped us improve our manuscript.

[GENERAL CONCERNS]

1. Discovery of a new fusion gene, TFG-RET, was not confirmed by sequencing and its functional role in PTC was not clear.

The fusion was detected by RNA-seq. Fusions are not typically detectable by exome data as most break points happen in the intronic regions (as was the case here). We have now PCR-amplified the cDNA of the patient sample (primary cancer) with primers that amplify the fusion regions and sequenced the PCR product. These data are now added to the supplement (figure S2). We are unable to perform any further experiments as we do not have any more of the patient RNA, DNA or cDNA.

We employed several studies, including *in vivo* studies (new figure 2E-G) to investigate the functional role of TFG-RET. We established human thyroid follicular epithelial cell line Nthy-ori-3 that stably expressed TFG-RET (as PTC arises from the follicular epithelial cells of thyroid [1]) to study the functional role of TFG-RET. Stable expression of TFG-RET lead to increased viability (MTT assay – figure 2A), increased cell proliferation (EdU assay – figure 2B), increasing oncogenic signalling, (figure 2D, S3A) transformation in soft agar (figure 2C) as

well as tumor formation *in vivo* (figure 2E-G). Our *in vivo* studies clearly showed pathologically visible tumors in mice injected with Nthy TFG-RET cells after a latency of about 12 weeks (figure 2E). Taken together, these data clearly confirm that TFG-RET is a novel thyroid oncogene.

2. “metastasis” was used for LN metastasis. Since metastasis includes LN and distant metastasis, it was better to use LN metastasis to distinguish from distant metastasis.

Thank you very much for pointing this out. We have now used the expression ‘LN metastasis’ throughout the manuscript where appropriate.

3. Some description was too short to explain data, including Results, Materials and Methods, and Figure legends, and it was difficult to follow.

In the revised version we have modified the text to better explain the data as well as materials and methods.

4. Use the word consistently. For example, tumour and tumor, and 100 mm and 10 cm, were used in this manuscript.

We have now modified the text for consistency.

5. Company information was not completed in many section in Materials and Methods.

We have now added additional company information in the materials and methods section. We have also completed and submitted the reporting summary as per the requirement of Nature Communications which included details of materials and analyses.

[SPECIFIC CONCERNS]

6. Introduction: Between Line 61 and Line 80 was too long to explain other oncogene in PTC. Instead, it is better to review RET fusion protein more.

Thank you for the suggestion. We have now more elaborately explained RET fusions in the introduction. We included the following text.

The *BRAF* mutation resulting in its highly kinase active protein form BRAFV600E and mutations in the *RAS* gene, especially *NRAS*, are the most common genetic alterations seen in PTC [7]. Apart from *BRAF* and *NRAS* mutations, common genetic alterations in PTCs include gene fusions involving the *RET* gene giving rise to oncogenic fusion proteins that account for up to 13-25% of PTCs [7, 12]. While *BRAF* mutations are prevalent in older patients, *RET* fusions are much more prevalent in younger patients. *RET* fusions (also called *RET/PTC* rearrangements) are genomic rearrangements that

are associated with ionising radiation-induced DNA damage. RET rearrangements were reported in up to 60% cases of post Chernobyl PTCs [13]. Spatial contiguity of the genes involved in the fusion during interphase could be the structural basis of these chromosomal rearrangements [14]. In oncogenic RET rearrangements the kinase domain-containing C-terminus of the RET gene, which is normally not expressed in thyroid follicular cells, is fused to the promoter-containing N-terminus of a ubiquitously expressed, unrelated gene [15].

7. Line 96: What is SOP? Standard operation procedure? Please describe specific procedure.

Yes, we meant standard operation procedure. We have detailed the procedure in the methods section.

8. Line 100: Exome data give us gene information, not protein. Please use the word "14 gene" instead of "14 protein". This applies to Figures and Figure legends.

We have used 14 protein-altering gene mutations for better clarity.

9. Figures S1A and S1B, and Table 1: The Reviewer did not understand figure S1B and Table 1. Tumor had 16 and metastasis had 9 fusion gene? How these genes were listed in Table 1? Table 1 showed 13 in tumor and 7 in LN. Please explain more clearly either in Results or legends.

We thank the reviewer for his/her feedback and have updated the legend describing Figure S1A and B. For the plot showing the number of protein-altering mutations, ALCAM and PPP1CC have 3 and 2 mutations, respectively - hence 16 mutations in tumor and 9 in metastasis. We have made this more obvious in table 1 by adding a new column, total mutations in the genes.

10. TFG-RET fusion was identified by RNA-seq analysis. Did the Authors confirmed this fusion gene by sequencing of gene from primary cancer and LN metastasis?

Yes! This was detected by RNA-seq data for both the primary and the metastasis sample (and not in the matched normal sample) and we had multiple reads supporting the fusion. RET fusions are well documented as drivers in papillary thyroid cancers (see TCGA). RET overexpression was consistent with known RET fusion biology and no other likely drivers were detected.

Additionally, we have now PCR-amplified the c-DNA of the patient sample (primary cancer) with primers that amplify the fusion region and sequenced the PCR product (figure S2). No further experimental validation was undertaken as we do not have any more of the patient RNA, DNA or c-DNA.

11. Supplementary table 1: Table showed 1734 genes. It was better to make another table including only 244 genes.

This new table has been included as Supplementary table 2.

12. Figure C: Figure lacks a labelling of primary and LN. It was not sure whether RET overexpression was due to TFG-RET. Did the authors examine other RET fusion genes and RET gene?

The figure is now labelled with 'normal', 'tumor' and 'metastasis'.

We examined previously described RET fusions in papillary thyroid cancers [1] and found that RET fusions lead to significant overexpression of the RET gene (Please see the new figure, Fig S1G, p-value is < .00001).

13. Line 113: Stably transfected NThyOri3 cells were pooled cells or single cell-derived clone? Please clarify them. If single clone was used, at least one more clone should be tested.

In order to avoid clonal artefacts, stable cell lines derived from pooled cells were employed in the presented analysis – as mentioned in materials and methods. Western blotting for the detection of the FLAG tag confirmed incorporation of TFG-RET in these cells. These cells were employed in transformation assays as well as *in vivo* studies. We have consistently observed the oncogenicity of TFG-RET-expressing cells, while the empty vector control cells did not induce any oncogenicity.

14. Figure 2A: Since MTT assay is cell viability assay, not cell proliferation assay, data from different day should not be compared. It is possible to be affected by cell density, medium pH, and other factors. Please confirm this data by counting cell number.

These comments are well taken. We have performed EdU DNA synthesis assay to confirm the cell proliferation data. Our data shows that stable expression of TFG-RET leads to a significant increase in cell proliferation when compared to empty vector control (new figure 2B).

15. Figures 3C and 3D: Figure legend for 3C was wrong. Please explain the number at the top of figure. If the molecular size TFG-RET protein were 75 kDa, why were they eluted much earlier? Please explain this.

Thank you very much for pointing this out. The figure legends have been corrected. These are presented as new figures 4C and 4D

Altered elution positions and undesirable chromatographic resolution as a consequence of protein adsorption to the column matrix are inherent drawbacks of size exclusion chromatography [2]. Hence, the proteins might have been retained in the column longer, hence eluting later.

The protein size from the samples on the western blot of the collected fractions of gel filtration confirms that the size of TFG-RET is around 75kDa (figure 4C). Further, the cross-linking studies using DTME, followed by disruption of the crosslinked heteromeric complex using DTT corroborates that the size of monomeric TFG-RET is around 75kDa (figure 3C).

16. Figure S3A: It was important to show some downstream protein changes, such as pERK and pAkt in addition to pRet to support functionally activating.

We have included additional data (figure S4C). This data shows that the increase in phosphorylation of STAT3 (a RET phosphorylation substrate) and MEK1/2 due to expression of TFG-RET is reduced in cells expressing TFG-RET Δ 97-124 and TFG-RETK14ER22ER23E when compared to cells expressing TFG-RET, despite the mutant TFG-RET exhibiting higher expression compared to TFG-RET.

17. Line 141: Fig. S2E must be Fig S3C?

We have corrected the figure reference.

18. Line 153: Please change the word, “deregulated” to more specific word, “increased” or “decreased”.

We have now changed the text as follows:

Interestingly, we found that several members of the ubiquitin signalling machinery are upregulated in the tumor and the LN metastatic lesion (Fig S5A).

19. Line 172: In thyroid cancers, many fusion proteins, including RET, BRAF, ALK, NTRK, PPARgamma, were reported and characterized. This sentence made confusion, and gave different message.

This has been modified as follows:

The discovery of oncogenic gene fusions including BCR-ABL and ALK fusions has led to the development of successful targeted therapies, particularly in liquid cancers [27]. With the recent advances in next generation sequencing techniques, there has been a massive increase in the number of molecular fusions described, especially in solid tumors. A TCGA study that aimed at the genomic characterisation of 496 PTCs illustrated the mutual exclusivity of genetic driver alterations in PTCs which further emphasizes the importance of precision medicine in the treatment of cancer [16].

20. Line 179: Reference 26 listed only 12 RET fusion proteins. Please cite right reference, which showed more than 20 RET fusion proteins.

Thank you for pointing this. We have added the following additional references.

de Groot, J.W., et al., RET as a diagnostic and therapeutic target in sporadic and hereditary endocrine tumors. *Endocr Rev*, 2006. 27(5): p. 535-60.

Julia Isabelle Staubitz, T.J.M., Arno Schad, Erik Springer, Hauke Lang, Krishnaraj Rajalingam, Wilfried Roth, Nils Hartmann, ANKRD26-RET - A novel gene fusion involving RET in papillary thyroid carcinoma. *Cancer Genetics*, 2019. 238: p. 10-17.

de Groot, J.W., et al., RET as a diagnostic and therapeutic target in sporadic and hereditary endocrine tumors. *Endocr Rev*, 2006. 27(5): p. 535-60.

21. Patient: Consent was described, but institutional approval was not clearly described. What was the surgical procedure? Total thyroidectomy and neck LN dissection? Please specify the surgery.

The procedure involved a total thyroidectomy with therapeutic central and right lateral lymph node dissection (neck dissection, levels II – VII). The lymph node was chosen because the macroscopic appearance was clearly metastatic.

Histology described a multifocal, bilateral follicular differentiated papillary thyroid carcinoma (pT4b, pN1, M1(lung), Stage II, Classification according to 1997).

We have ethical approval for the study granted by the “Ärztchamber” which is also the institutional approval (Study number: 837.119.15 (9888)).

This information has now been added to the manuscript in the methods section.

22. Line 229: The Authors obtained only one LN? Why this one LN was chosen?. Did they perform confirmatory experiments using DNA from other LNs?

The lymph node was chosen because the macroscopic appearance was clearly metastatic.

Histology described a multifocal, bilateral follicular differentiated papillary thyroid carcinoma (pT4b, pN1, M1(lung), Stage II, Classification according to 1997).

23. Line 258: There was other section describing antibodies. Please move to there or change the description. Ex. Antibodies against thyroglobulin and calcitonin were purchased from DAKO.....

We have now moved the details of thyroglobulin and calcitonin to the section containing the details of antibodies.

24. *Invitrogen and Thermo Fisher are the same company in the US. Please use Thermo Fisher.*

We have used Thermo Fisher.

25. *Line 271: Please explain PEI.*

PEI is polyethylenimine. We have incorporated the explanation.

26. *Line 273: What were experiments carried out 48 hours post transfection. From this sentence, it was not clear.*

This information is now incorporated in the respective sections.

27. *Line 284: How were gene expression normalized? Please explain.*

Expression was normalised using reference genes (18S or RSP13). This has been now added in text and the primer sequences were added to the methods section. These genes have been validated and routinely used as reference genes in our lab.

28. *Line 298: Missing the information of antibody A7906 from Sigma.*

A7906 refers to the product number of BSA from Sigma. This is now clarified.

29. *Line 328, Cell proliferation assay: MTT is not the proliferation assay. This is cell viability assay.*

We court this correction and it is well taken. In response we have performed EdU assay as a measure of cell proliferation (figure 2B). The EdU assay showed an significant increase in cell proliferation in cells expressing TFG-RET.

30. *Line 344: Replace to with on.*

We changed the sentence accordingly.

31. *Line 348: Please add the source of Image J.*

Below is the source of Image J. It has been added into the main text.

Open source image processing software <http://imagej.net, Version: 2.0.0-rc-69/1.52i>.

We thank Reviewer #1 for his substantial comments. We believe that we have been able to address the comments to the fullest.

Reviewer #2 (Expertise: DUBs, Proteomics, Remarks to the Author):

The authors present the analysis of a single thyroid tumour of a patient with TFG-RET fusion. They show that the TFG-RET oncogene oligomerises and this is important for transformation. Proteomics analysis of the tumour shows up-regulation of ubiquitin ligase HUWE1 as well as DUBs Usp9X and UBP7. Over-expression of TFG-RET in cell culture leads to up-regulation of HUWE1 and pharmacological inhibition of RET. HUWE1 and DUBs reduced oncogenesis.

We thank the reviewer for his/her time, comments and views. We take the comments and suggestions very positively and have made every effort to address all the concerns raised.

Major points.

• The authors' main problem is that they have n=1. Proteomics and transcriptomics analyses with one bioreplicate are rather problematic.

Well taken! However, we would like to emphasize that with individualized medicine and therapeutics, every patient needs to be analysed in detail to design tailor-made, rational therapeutics.

Nevertheless, we have now added data analysing the expression of HUWE1, USP7 and USP9X in a larger cohort of duly consented PTC patients (figure 5D, E). We would like to bring it to the kind attention of the reviewer that we performed these studies on fresh frozen materials (matching normal, tumour and metastatic tissue) which are valuable samples donated by patients. It has taken quite some time to muster a significant number of patient-derived materials to perform these analyses.

The samples were used to compare the expression of HUWE1, USP7 and USP9 between normal and tumor samples. The data was quantified, and the results showed that these proteins tend to have higher expression in many tumors and metastatic tissues.

• Figures are not well described. It would be much better if the figure panels included small titles such as "MTT assay" or "proliferation" (Figure 2).

Thank you for the suggestion. We have now added titles within the figures, wherever possible.

• For the proliferation assays, the over-expression of the WT RET is missing. It is quite obvious that over-expressing an oncogene leads to more proliferation, also when the oncogene is fused to TFG.

We have now performed EdU DNA synthesis assay where WT RET overexpressing cells were included (Figs 2D, S3A). Expression of WT RET

showed a slight increase (not significant) in cell proliferation, while TFG-RET overexpression lead to a significant increase in cell proliferation. WT RET upon overexpression per se can be activated by dimerization which could account for the slight increase in proliferation.

Further, we would like to emphasise that there is an increased expression of RET in PTCs harboring RET fusions compared to PTCs without fusion RET (please refer figure S1G).

• *Some of the data is just not good enough to draw the conclusions that the authors have drawn. For example, Figure S4B appears to present three replicates of cell cultured cells overexpressing TFG-RET. The loading control vinculin is off, USP7 is all over the place and none of the proteins HUWE1, USP7 and USP9X really changes reliably.*

We have now analysed the expression of HUWE1, USP7 and USP9X in a larger cohort of patient samples (normal vs tumor and metastasis) and also quantified the blots, which indicated that these proteins are up-regulated in the tumor and metastatic lesions (figure 5D). We believe that these results further strengthen our observation that HUWE1, USP9X and USP7 are indeed stabilised in PTC patients.

• *The paper is rather too concisely written and thus, one seems to miss details. The methods are too short for anyone to reproduce. Are the replicates in proteomics technical replicates and at which level? Reinjection of the same sample? Details of the inhibitor treatment is missing in methods.*

We have done extensive revision of the draft to ensure that the paper is more detail and clearer.

The replicates in the proteomics data are technical replicates, and the same sample was used for 3 different injections.

The details of inhibitor treatments are now presented in the methods.

• *The volcano plot in figure indicates non-equal loading of samples as everything is up in the tumour. How did the authors normalise?*

We thank Reviewer 2 for this comment. For label-free quantification, the data were normalized based on total ion current (TIC) in the PEAKS software to correct for unequal sample loading.

We agree that the observed asymmetrical regulation pattern is unusual. However, we interpret this as a sign of tissue de-differentiation, where several highly abundant (tissue-specific) proteins are downregulated, while a larger number of proteins is upregulated during tumorigenesis.

Additionally, our new observation in multiple patient samples (figure 5D) validated the candidate proteins that we obtained from the proteomics data.

• The DUB and HUWE1 inhibitors are unspecific. For WP1130 you can look here: PMID: 25159004. There is a published USP7 inhibitor that is specific and you can find the paper here: PMID: 29045385. It is not surprising that these unspecific inhibitors kill the cells. You would need to compare treated cells overexpressing RET-TFG versus plasmid control and overexpressing RET alone.

Comment well taken. In order to address the specificity of inhibitors, we have used siRNAs to knockdown HUWE1, USP7 and USP9X (figure 6A) and conducted viability assays (MTT – Figure 6B), proliferation assays (figure S6A, B) and soft agar colony formation assays (figure 6C, S6D).

We would not include plasmid control Nthy-Ori-3, as our data show that these cells could not form colonies (figure 2C).

• shRNA experiments of these enzymes will be more reliable.

We very much agree with the comment, and we have tried hard to establish shRNA knockdown cell lines for conducting our studies. Yet despite our best efforts, we were unable to establish cell lines that stably expressed TFG-RET and had a stable shRNA knockdown. Please note that there are no commercially available PTC cell lines. Here, we have transformed Nthy cells with TFG-RET and used those cells in all these experiments. It is not a classical tumour cell line which is amenable for shRNA transduction and selections. Hence, we used a transient knockdown approach using siRNAs to functionally validate the role of HUWE1, USP7 and USP9X in NTh-Ori-3 cells stably expressing TFG-RET.

As mentioned above, we carried out cell viability assay (MTT assay – figure 6B), cell proliferation experiments (EdU assay, cell counting - figure S6A-C) as well as soft agar colony formation assay (figure 5C, S6D). The results indicated potential roles for some of the proteins in cell proliferation, viability and oncogenicity. All these data have been added to the revised version of the manuscript.

Of course, these experiments need further validation. But we believe that our study calls for investigation of HUWE1, USP7 and USP9X as potential therapeutic targets in cancers with RET fusions, which might subsequently open novel avenues for therapeutic strategies.

We would like to again thank Reviewer #2 for his/her comments. We believe that addressing these concerns helped us to strengthen our observations.

Reviewer #3 (Expertise: Gene fusions, RNA sequencing, Remarks to the Author):

Krishnan et al performed exome and transcriptome sequencing on a triplet of a thyroid tumor, a metastatic lesion, and an adjacent normal sample. A

RET fusion was identified that has not been previously described, and using several functional approaches this fusion was found to transform immortalized cells. Small molecule inhibitors of RET and downstream proteins were shown to reduce RET-mediated oncogenesis. RET fusions have recently gathered more interest due the favorable responses seen in patients with RET-fusion positive tumors after treatment with for example LOXO-292.

We thank Reviewer #3 for his/her observations and comments. We have made our best efforts to address the concerns put forth, and we hope that this has helped in putting more context to the work.

Addressing the following questions would help to provide context to this work.

- 1. TFG fusions are often caused by a germline copy number variant. Can the fusion be validated genomically, and be shown to be unique to the tumor?***

RNA sequencing was the method used for the identification of TFG-RET fusion. The fusion was detected in both the primary and the metastasis sample (and not in the matched normal sample) and we had multiple reads supporting the fusion.

Fusions are not typically detectable by exome data as most break points happen in the intronic regions (as was the case here). Genomic sequencing of the DNA would be challenging due to the larger intronic inserts.

However, we have now PCR-amplified the cDNA of the patient sample (primary cancer) using primers for the fusion region (figure S2) and sequenced the PCR product. We are unable to perform any further experiments as we do not have any more of the patient RNA, DNA or c-DNA.

- 2. Resources such as tumorfusions.org and ChimerDB have enumerated fusions across many cancers. How often is the same TFG-RET fusion seen in other tumors?***

To our knowledge, the TFG-RET fusion has not been reported anywhere so far.

Below is our search result from tumorfusions.org

Gene-Centric Fusions Results											
Gene Name: <input type="text" value="sora"/> Query: eg:EGFR											
Gene constituting a part of fusion can be individually queried by Gene Symbol here.											
Show to 3 entries										Search:	
Cancer	TCGA SampleId	FusionPair	E-value	Tier	Frame	5' Gene Junction	3' Gene Junction	Details	CNV	Somatic Muts	WCS
THCA	TCGA.FK.A353.01A	MET_TFG	1.6	tier1	In-frame	Chr7:116412043/1	Chr3:100455420/1		MET TFG	MET TFG	validated
THCA	TCGA.FK.A353.01A	TFG_MET	1.6	tier1	In-frame	Chr3:100451516/1	Chr7:116414935/1		TFG MET	TFG MET	validated
THCA	TCGA.EM.A3A0.01A	TFG_NTRK1	100.0	tier1	In-frame	Chr3:100455560/1	Chr1:156844363/1		TFG NTRK1	TFG NTRK1	NA

Showing 1 to 3 of 3 entries First Previous 1 Next Last

3. What is the expression of the fusion transcript relative to full RET transcript.

We examined previously described RET fusions in papillary thyroid cancers (1) and found that RET fusions lead to significant over expression of the RET gene. This information is now included in the manuscript (figure S1G).

4. Fig. 1C. Legend refers to exome analysis as well as overexpression. What does this figure represent, exome sequencing does not measure gene transcription.

The fusion was detected from RNA-seq data (NOT Exome data). We have changed the figure legends and added more information for further clarity (figure S1).

We would like to thank Reviewer #3 for the comments, and we believe that we have been able to answer all the questions.

Overall, we have addressed the concerns of all the reviewers in full, and we sincerely hope that the reviewers support the publication of this manuscript in nature communications.

1. Gimm, O., Thyroid cancer. *Cancer Lett*, 2001. 163(2): p. 143-56.
2. 1. Cancer Genome Atlas Research, N., *Integrated genomic characterization of papillary thyroid carcinoma*. *Cell*, 2014. 159(3): p. 676-90.
3. 2. Arakawa, T., et al., *The critical role of mobile phase composition in size exclusion chromatography of protein pharmaceuticals*. *J Pharm Sci*, 2010. 99(4): p. 1674-92.

Reviewers' comments:

Reviewer #1 (Expertise: Papillary Thyroid cancer, metastasis, Remarks to the Author):

The Authors responded and addressed to most of critiques. There are still minor correction, for example,

LN is not explained in line 94.

Figure S3C and D needs to explain what is band of 17 kDa and >171 kDa (TFG-RET) and why the Authors showed.

Figure 6D, It was hard to see colonies on figures.

Reviewer #2 (Expertise: DUBs, proteomics, Remarks to the Author):

I thank the authors for inclusion of some of my points. However, I still feel that details are missing, the proteomics data on a single replicate (even if 3 technical injections) is meaningless, particularly since you have a huge shift to the right, indicating unequal loading.

- Please deposit the proteomics data in PRIDE or a similar repository – as all should be doing now.

- Secondly, please clarify the figure legend in Fig 5 more. For example, when you say in E, the expression of X, Y,Z was quantified, please explain how you did that. Was that from Western blots?

- Figure 6: the MTT assay. It looks like the authors normalised the control thereby getting a variance of 0, which will improve their statistical tests massively. You cannot do that. Moreover, this normalisation is nowhere described. Please use absolute numbers in these assays.

- The volcano plot in figure 5 appears to be from 3 technical replicates. why is this nowhere written? How shall anyone ever reproduce anything if this information is missing? You cannot make volcano plots of technical replicates. This just measures your technical variance but makes no biological sense.

Reviewer #3 (Expertise: Gene fusions, RNA sequencing, Remarks to the Author):

In the resubmission by Krishnan et al, the key question that is not addressed is the validity of the fusion that initiated the study. There is no evidence to demonstrate that this fusion is a bona fide somatic tumor driving event, other than that the authors claim that the fusion exists in this tumor sample.

1. A comment from the initial paper was to demonstrate that the fusion is a somatic and not a germline event. This question was not addressed as the authors state to not have any germline material remaining. This is unfortunate as TFG fusions are frequently caused by a germline copy number variant. Note that sequencing is not the only approach able to validate the fusion - cytogenetic approaches should be able to demonstrate a chr 3- chr 10 translocation. Without this evidence, it is not proven that the fusion is a tumor-specific event. The absence of RET gene expression in normal thyroid tissue does not change that.
2. Another comment from the original submission was how the transcript levels of the fusion transcript compares to the transcript levels of the full length transcript. In response, Figure S1G showing an analysis of TCGA data was newly included but this is not answering the question asked. The cDNA analysis of Figure S2 provides some suggestion that the gene expression of the fusion product is not as high as might expected in the context of the RET expression figure S1E. Showing that the tumor/met express the fusion gene and not the wild type RET transcript would provide evidence that the RET fusion is functionally relevant. Please compare the number of RNAseq read pairs with one read mapping to TFG and the other to RET , or TFG-RET junction spanning RNAseq reads, to the number of RET reads.
3. Additionally, providing evidence of the fusion by showing the number of RNAseq read pairs that map to TFG and RET, and/or the number of RNAseq sequence reads that span the TFG-RET junction, would help. Showing a TFG-RET exon expression plot would help as well.

We thank the reviewers for the additional concerns which are now fully addressed. Please find our responses below.

Reviewer #1 (Expertise: Papillary Thyroid cancer, metastasis, Remarks to the Author):

The Authors responded and addressed to most of critiques. There are still minor correction, for example,

LN is not explained in line 94.

Thanks for pointing this out. This has been corrected.

Figure S3C and D needs to explain what is band of 17 kDa and >171 kDa (TFG-RET) and why the Authors showed.

Sorry for not being clear here. LE- stands for the „longer exposure“ of the same blots shown above. It is 171 KDa and not 17 KDa. This has been corrected in the figure but also mentioned in the legends.

Figure 6D, It was hard to see colonies on figures.

We have removed the figure with the colonies as the resolution is not high enough. However, the quantification says it all: Knock down of HUWE1 led to a reduction in colony formation of TFG-RET expressing cells.

Reviewer #2 (Expertise: DUBs, proteomics, Remarks to the Author):

I thank the authors for inclusion of some of my points. However, I still feel that details are missing, the proteomics data on a single replicate (even if 3 technical injections) is meaningless, particularly since you have a huge shift to the right, indicating unequal loading.

- Please deposit the proteomics data in PRIDE or a similar repository – as all should be doing now.

We followed the advice of this reviewer and the data are now deposited in PRIDE . Please find the links below :

Project Name: Proteogenomics analysis unveils a novel TFG-RET gene fusion and druggable targets in papillary thyroid carcinomas
Project accession: PXD016739 and PXD016828

Reviewer account details:

Username: reviewer51048@ebi.ac.uk
Password: X8d805PB

For the data on the patient with TFG-RET fusion
Username: reviewer35877@ebi.ac.uk
Password: 5Xk7zgvZ

- Secondly, please clarify the figure legend in Fig 5 more. For example, when you say in E, the expression of X, Y,Z was quantified, please explain how you did that. Was that from Western blots?

Protein bands and thus the expression were quantified from the blots with Image J (now presented as Figure S5) . Ratio between the selected protein of interest to GAPDH in tumor or metastasis was normalized to normal tissues. For those where no normal tissue was available (patient 7&10) the mean of all normal tissues (n=7) was used for normalization. Significance was determined with a paired t-test. This methodology is now explained in the legends as well as in the methods section.

- Figure 6: the MTT assay. It looks like the authors normalised the control thereby getting a variance of 0, which will improve their statistical tests massively. You cannot do that. Moreover, this normalisation is nowhere described. Please use absolute numbers in these assays.

MTT assay was recalculated throughout the manuscript according to standard procedures. The absolute OD values measured are now converted into percentage of viable cells and shown in Figs. 6 and S6. In short, 4 independent experiments (N=4;biological replicates) were analyzed; for each experiment and for each condition the average of three technical replicates was calculated. Then the percentage of viable cells was determined using this formula: Percentage of viable cells = $100 * (\text{OD of siRNA transfected cells} - \text{OD of the background}) / (\text{OD of siControl transfected cells} - \text{OD background})$. The methods section has been expanded accordingly to make it more vivid.

- The volcano plot in figure 5 appears to be from 3 technical replicates. why is this nowhere written? How shall anyone ever reproduce anything if this information is missing? You cannot make volcano plots of technical replicates. This just measures your technical variance but makes no biological sense.

Following the advice of the reviewer, we have removed the displayed volcano plot from the main figures, as we agree with the reviewer that a volcano plot of technical replicates has a somewhat limited value as it will display only the technical variability of the proteomic workflow (in this case, covering nanoLC separation, MS analysis and data processing/evaluation).

However, as we think that the reproducibility of the workflow is a critical aspect to enable the reader to judge the validity of the data presented, we opted to display the volcano plots of the newly analyzed samples in Suppl. Figure S5C.

We have now expanded our proteomic analysis with 4 more patients and these new data are now presented to strengthen our claims as a main figure (please see new Figs. 5B, C and the supplement table 5). The respective raw data have also been deposited in the PRIDE repository as advised.

While the expression levels of HUWE1 were different between the individual patients, we detected HUWE1 as one of the prime targets being upregulated in the tumour and metastatic tissue in three out of four of the newly analyzed patients (Please see Fig. 5B). We have also identified other interesting targets like KRAS which has also been validated and added to the manuscript (Fig. 5C). Overall, we are confident that the new dataset have significantly strengthened our study substantially.

Reviewer #3 (*Expertise: Gene fusions, RNA sequencing, Remarks to the Author*):

In the resubmission by Krishnan et al, the key question that is not addressed is the validity of the fusion that initiated the study. There is no evidence to demonstrate that this fusion is a bona fide somatic tumor driving event, other than that the authors claim that the fusion exists in this tumor sample.

1. A comment from the initial paper was to demonstrate that the fusion is a somatic and not a germline event. This question was not addressed as the authors state to not have any germline material remaining. This is unfortunate as TFG fusions are frequently caused by a germline copy number variant. Note that sequencing is not the only approach able to validate the fusion - cytogenetic approaches should be able to demonstrate a chr 3- chr 10 translocation. Without this evidence, it is not proven that the fusion is a tumor-specific event. The absence of RET gene expression in normal thyroid tissue does not change that.

>> We have looked at presence of fusion transcript in the tumor, metastasis, patient matched normal using RNA-seq data and subsequently with Sanger sequencing of the cDNA. We find evidence of the fusion in the tumor and the metastatic sample, but not in the matched normal confirming that this is a somatic event and not a germline event (Also see new Figs Supplementary Fig. S1D and S2). We have

identified the fusion junction reads and found evidence for these only in the tumor and metastatic sample, further confirming the somatic and not a germline event. While the cytogenetic approach can validate the fusion, sequencing based approaches are accurate and provide a more direct evidence and now being applied in the clinic. Also, note that while this paper was under review a study reported the presence of exon 8 – exon 12 TFG-RET fusion in a child with spindle cell cancer (Loong et al., 2019). The fusion we identified here is an exon4 – exon11 TFG-RET fusion. Further occurrence of RET fusions are common in PTCs.

2. Another comment from the original submission was how the transcript levels of the fusion transcript compares to the transcript levels of the full length transcript. In response, Figure S1G showing an analysis of TCGA data was newly included but this is not answering the question asked. The cDNA analysis of Figure S2 provides some suggestion that the gene expression of the fusion product is not as high as might expected in the context of the RET expression figure S1E. Showing that the tumor/met express the fusion gene and not the wild type RET transcript would provide evidence that the RET fusion is functionally relevant. Please compare the number of RNAseq read pairs with one read mapping to TFG and the other to RET , or TFG-RET junction spanning RNAseq reads, to the number of RET reads.

>> We show the reads for TFG, RET and TFG-RET fusion junction in the revised manuscript (see Fig S1D, Also find attached the excel table showing the actual reads Supplementary table 1). We find evidence for TFG-RET in tumor and metastatic sample but not in the matched adjacent normal. Further the levels of fusion junction transcript evidence increased in the metastatic tumor tissue suggesting a selection and enrichment for TFG-RET mutant cells.

3. Additionally, providing evidence of the fusion by showing the number of RNAseq read pairs that map to TFG and RET, and/or the number of RNAseq sequence reads that span the TFG-RET junction, would help. Showing a TFG-RET exon expression plot would help as well.

>> We show the reads for TFG, RET and TFG-RET fusion junction in the revised manuscript (see Fig S1D and supplementary table 1). Expression of RET past the fusion junction Exon 10) is elevated in tumor compared to the normal and is further increased in metastatic sample. Also, the expression of TFG is higher in the normal and fusing with it to drive the RET oncogene is indicative of a driver role for the fusion. We have indeed confirmed the oncogenic ability of this fusion by employing a variety of “transformation” assays including proliferation, anchorage independent growth (soft agar) by employing TFG-RET expressing cells which also formed tumours in immunodeficient mice.

Loong, S., D.W.Q. Lian, C.H. Kuick, T.H. Lim, S.A. Nah, K.P.L. Wong, and K.T.E. Chang. 2019. Novel TFG-RET fusion in a spindle cell tumour with S100 and CD34 coexpression. *Histopathology*.

REVIEWERS' COMMENTS:

Reviewer#2: (Remarks to the Author)

I thank the authors for including my comments.

I have one final wish. Please tone down on the use of the DUB and E3 ligase inhibitors. these are certainly unspecific as there are no tools at the moment to test the specificity of E3 ligase inhibitors against a large number of enzymes. As for the DUB inhibitor WP1130, this has been shown to be unspecific (Ritorto et al, Nature Comm 2014).

Please just write that these are tool compounds that are likely not very specific.

thanks.

Reviewer#3: (Remarks to the Author)

With the inclusion of the new Table S1, my final comments have been addressed.

Reviewer#2: (Remarks to the Author)

I thank the authors for including my comments.

I have one final wish. Please tone down on the use of the DUB and E3 ligase inhibitors. these are certainly unspecific as there are no tools at the moment to test the specificity of E3 ligase inhibitors against a large number of enzymes. As for the DUB inhibitor WP1130, this has been shown to be unspecific (Ritorto et al, Nature Comm 2014).

Please just write that these are tool compounds that are likely not very specific.

We have modified the respective text to align with the reviewers suggestions.